# MOCK: an Algorithm for Learning Nonparametric Differential Equations via Multivariate Occupation Kernel Functions

**Victor Rielly**  *victor23@pdx.edu*
*Department of Mathematics + Statistics*
*Portland State University*

**Kamel Lahouel**  *klahouel@tgen.org*
*Translational Genomics Research Institute*

**Ethan Lew**  *ethanlew16@gmail.com*

**Nicholas Fisher**  *nicholfi@pdx.edu*
*Department of Mathematics + Statistics*
*Portland State University*

**Vicky Haney**  *vhaney@pdx.edu*
*Department of Mathematics + Statistics*
*Portland State University*

**Michael Wells**  *mlwells@pdx.edu*
*Department of Mathematics + Statistics*
*Portland State University*

**Bruno Jedynak**  *bjedyna2@pdx.edu*
*Department of Mathematics + Statistics*
*Portland State University*

**Reviewed on OpenReview:** *https://openreview.net/forum?id=fjVIp2Z9RS&noteId=oKGpw1Z5d7*

## Abstract

Learning a nonparametric system of ordinary differential equations from trajectories in a $d$-dimensional state space requires learning $d$ functions of $d$ variables. Explicit formulations often scale quadratically in $d$ unless additional knowledge about system properties, such as sparsity and symmetries, is available. In this work, we propose a linear approach, the multivariate occupation kernel method (MOCK), using the implicit formulation provided by vector-valued reproducing kernel Hilbert spaces. The solution for the vector field relies on multivariate occupation kernel functions associated with the trajectories and scales linearly with the dimension of the state space. We validate through experiments on a variety of simulated and real datasets ranging from 2 to 1024 dimensions, and provide an example with a divergence-free vector field. MOCK outperforms all other comparators on 3 of the 9 datasets on full trajectory prediction and 4 out of the 9 datasets on next-point prediction.

## 1 Introduction

### 1.1 Description of the problem

The task of learning dynamical systems derived from ordinary differential equations (ODEs) has garnered a lot of interest in the past couple of decades. In this framework, we are often provided with noisy data from trajectories representative of the dynamics we wish to learn, and we propose a candidate model to describe

these dynamics. We present a learning algorithm for the vector field guiding the dynamical system that scales linearly with the dimensionality of the dynamical system's state space, allowing us to efficiently learn high-dimensional systems.

## 1.2 High Dimensional Dynamical Systems

Large systems of ODEs may arise from the numerical discretization of a PDE. They also arise in modeling the temporal evolution of complex systems composed of many interacting components. These systems appear across many disciplines. We provide some examples.

1. **The Method of Lines:** Consider a PDE of the form

$$\frac{\partial}{\partial t} u(x, t) = \mathcal{L} u(x, t) \tag{1}$$

   where $\mathcal{L}$ is a differential operator acting on the spatial variable(s). Examples include Burgers' equation, the heat equation, Fisher's equation, and other reaction-diffusion systems. The method of lines (Schiesser, 2012) is a technique for solving PDEs of the form (1) by discretizing the equation in all but the time variable, reducing it to a system of coupled ODEs (or, possibly, differential algebraic equations). The method proceeds as follows: upon discretizing this system by $U_i \approx u(X_i, t), i = 1 \ldots d$, we obtain the family of ODEs $dU/dt = G(U)$ where $U$ is a vector of dimension $d$ and $G$ is a $\mathbb{R}^d \to \mathbb{R}^d$ (possibly nonlinear) vector field. If the discretization has, say, 100 points along each of 3 spatial dimensions, this discretization leads to a system of ODEs of dimension one million.

   Note that the specific form of $G$ depends on the underlying dynamics and on the spatial discretization scheme. In this paper, we only assume that $G$ belongs to a vvRKHS, but we do not assume a specific parametric form, which may be important if $G$ is not fully specified.

2. **Chemical Reaction Networks and Agent-Based Models:** In biochemical systems, each species (a protein, ion, or small molecule) may follow a distinct concentration trajectory governed by mass-action kinetics or Michaelis-Menten dynamics. The average behavior often results in a high-dimensional ODE system representing the time evolution of all species in the network. More broadly, agent-based models that describe individual entities (cells, molecules, or agents) interacting via prescribed rules can often be approximated by high-dimensional ODEs (Anderson & Kurtz (2015), Murray (2007), Chapter 6).

3. **Networks of Neurons**: Neural populations can be modeled as networks of coupled nonlinear oscillators, such as the FitzHugh-Nagumo FHN model for individual neuron dynamics. When modeling a large number of neurons with electrical or chemical coupling, one obtains a high-dimensional ODE system (Murray (2007), Chapter 7).

4. **Protein Accumulation in the Brain and Neurodegenerative Dynamics:** Models of protein aggregation and propagation in diseases like Alzheimer's or Parkinson's involve spatial-temporal accumulation and diffusion processes. When discretized in space (e.g., across brain regions in Positron Emission Tomography imaging studies), these systems lead to high-dimensional ODEs for tracking the progression of misfolded proteins (tau, beta-amyloid).

5. **Mechanical Systems with Multiple Degrees of Freedom (Lagrangian and Hamiltonian Formulations):** In robotics and biomechanics, systems often have many degrees of freedom. Each joint or actuator contributes to the state space. The dynamics of such systems are derived from Newtonian, Lagrangian, or Hamiltonian mechanics, yielding structured high-dimensional ODEs (Featherstone (2014), Chapter 7, Ivancevic & Ivancevic (2010)).

## 1.3 State of the art

We compare MOCK to various dynamical systems learning methods, including sparse identification of nonlinear dynamics, reduced order models, deep learning methods, and neural ODEs. These methods learn

system dynamics from trajectories in the state space with good accuracy, scale to tens, hundreds, or thousands of state dimensions, and are robust to noise. A detailed description of the comparators is presented in Section 4.4 and in Appendix G.

In (Heinonen et al., 2018) and (Hegde et al., 2022), the method presented parametrizes the estimated vector field for the dynamical system with a zero-mean Gaussian Process (GP) whose covariance is given by a positive-definite kernel. A grid of inducing points is formed where the vector field is learned directly, and the full vector field is then interpolated from these inducing points. The use of the grid of inducing points makes this model prohibitively expensive in large dimensions as the number of grid points required increases exponentially with dimension, and thus we have excluded it from our comparison.

### 1.4 Main contributions

We propose to use MOCK to learn dynamical systems. This method scales linearly with the number of dimensions of the state space. Building upon (Rosenfeld et al., 2019a), (Rosenfeld et al., 2019b), (Russo et al., 2021), we provide a derivation with the help of vector-valued Reproducing Kernel Hilbert spaces (vvRKHSs) which allows for a reduction in computational complexity in special cases and extensions to physics informed kernels. We also emphasize the simplicity of the algorithm and demonstrate competitive performance on high-dimensional data, as well as noisy data. Finally, in Section 5 we craft a vvRKHS for learning divergence-free vector fields, demonstrating the versatility of the MOCK method.

Rosenfeld and Russo extend the setting of ridge regression to the case of learning a vector field using snapshots of trajectories, as presented in the paper (Rosenfeld et al., 2019a) and the follow-up paper (Rosenfeld et al., 2024). In both papers, a finite basis set of vector-valued functions was used to learn the unknown slope field. The provided solution is equivalent to optimizing in a vvRKHS with an explicit kernel. Using an I-separable kernel, a special type of matrix-valued kernel to be defined below, we allow our parameter count to increase linearly with the state space dimension ($d$) while maintaining computational complexity that increases only linearly in $d$. In contrast, the model proposed by Rosenfeld and Russo requires computational complexity to increase cubically with the number of parameters of the learned model.

In addition, we generalize this work to arbitrary matrix-valued kernels. Note that we keep the least square setting of ridge regression while in (Rosenfeld et al., 2024), Rosenfeld and Russo used a more general weak formulation with Liouville operators. We create a benchmark and compare MOCK against the state-of-the-art algorithms. We also benchmark several kernels, including Gaussian, Laplace, Matérn, and random Fourier features. Lastly, in Section 5, we present an application where the vvRKHS is made of divergence-free kernels. Using simulated data, we present an experiment in which the use of a divergence-free kernel improved the recovery of the vector field compared to an ordinary kernel.

Without any restriction on the vvRKHS, learning a vector field with MOCK requires solving an $(Nd, Nd)$ system, where $N$ is the total number of snapshots and $d$ is the dimension of the state space. Improvements can be obtained when the kernel is a tensor product of univariate kernels. In this case, which includes I-separable kernels, one needs to solve $d$ $(N, N)$ linear systems. When the univariate kernel is explicit, and made of $q$ feature functions, one can instead solve $d$ $(q, q)$ linear systems. By contrast, Russo and Rosenfeld considered a loss equivalent to the case of an arbitrary explicit vvRKHS without the tensor product structure. This has the drawback of requiring a runtime that scales cubically with the total number of parameters in the model (requires solving a $dq \times dq$ linear system). This is also shared by SINDy and the explicit forms of DMD.

## 2 Background

### 2.1 vvRKHS for modeling vector fields

Scalar RKHSs are spaces of real-valued functions made popular for their use in constructing the kernelized support vector machine classifier (Schölkopf & Smola, 2002) chapter III. Scalar RKHSs generalize to vvRKHSs and provide simple nonparametric models for vector fields. A matrix-valued kernel fully characterizes a

vvRKHS. The choice of this kernel is left to the user and allows for encoding various properties of the functions in the corresponding vvRKHS. Choosing a kernel with Lipschitz continuous diagonal elements guarantees that all functions in the uniquely associated vvRKHS, denoted $\mathbb{H}$, are Lipschitz continuous. This, in turn, guarantees the local existence and uniqueness of the solutions to the associated ordinary differential equation $\dot{x} = f(x)$ where $f \in \mathbb{H}$, see (Lahouel et al., 2022). Furthermore, one may allow the inclusion of physics-inspired constraints into the function space by appropriately choosing the kernel, and an example of a divergence-free vvRKHS is provided in Section 5. Kernels come in two forms: implicit and explicit. The former implicitly characterizes a mapping from $\mathbb{R}^d$ to a Hilbert space, which can be of infinite dimension. Examples include the family of Matérn kernels, which contain the familiar Gaussian and Laplace kernels. The latter explicitly characterizes a mapping from $\mathbb{R}^d$ to $\mathbb{R}^p$ for some $p$. Examples include polynomial kernels and random Fourier features. Both kernel types will be used in the experiments in Section 4. We now provide a mathematical presentation of kernels and vvRKHSs.

## 2.2 Vector-Valued RKHS (vvRKHSs)

*Definition* 1. Let $\mathcal{M}_{(d,d)}$ be the set of $(d,d)$ matrices, $d \geq 1$. A positive definite matrix-valued kernel $K$ is a mapping: $\mathcal{X} \times \mathcal{X} \mapsto \mathcal{M}_{(d,d)}$, such that

1. symmetric: $K(x, x') = K(x', x)^T$, for any $x, x' \in \mathcal{X}$; $M^T$ denotes the transpose of the matrix $M$.

2. positive definite: for any $x_1, \dots, x_n$ in $\mathcal{X}$, and for any $w_1, \dots, w_n$ in $\mathbb{R}^d$,

$$\sum_{i,j} w_i^T K(x_i, x_j) w_j \geq 0 \tag{2}$$

The simplest kernels are the *separable* kernels

$$K(x, x') = k(x, x')A \tag{3}$$

where $k$ is a positive definite scalar kernel and $A$ is a positive semi-definite (PSD) matrix. When $A = I$, $\mathbb{H}$ is made of $d$ copies of the RKHS of $k$. We call such a kernel I-separable. If the kernel $k(x, x')$ is an explicit kernel (can be written as $\phi(x)^T \phi(x')$ for some function $\phi : \mathbb{R}^d \to \mathbb{R}^q$) it has the additional properties:

$$K(x, x') = \phi(x)^T \phi(x')I \tag{4}$$

$$= \left(\phi(x)^T \phi(x')\right) \otimes \left(I^T I\right) \tag{5}$$

$$= \left(\phi(x) \otimes I\right)^T \left(\phi(x') \otimes I\right) \tag{6}$$

where $\otimes$ denotes the Kronecker product. Defining $\Psi(x) : \mathbb{R}^d \to \mathbb{R}^{qd \times d}$ by

$$\Psi(x) = \phi(x) \otimes I, \tag{7}$$

we see that the matrix-valued explicit kernel is

$$K(x, x') = \Psi(x)^T \Psi(x') \tag{8}$$

All kernels used in our experiments in Section 4 are of the I-separable form, while the divergence-free kernel in Section 5 is not an I-separable kernel. There are three ways to characterize a vvRKHS. They are presented in Appendix D for completeness. Here, we briefly present the construction using the Riesz representation theorem, which is critical in developing the MOCK algorithm.

*Definition* 2. Let $\mathbb{H}$ be a Hilbert space of functions $\mathcal{X} \to \mathbb{R}^d$. We denote by $\langle \cdot, \cdot \rangle$ the inner product in $\mathbb{H}$ and $\|\cdot\|_H$ or simply $\|\cdot\|$ the associated norm. Critically, we assume that for any $x \in \mathcal{X}$, the evaluation functional $f \mapsto f(x)$ is continuous, that is, there is a constant $M_x \in \mathbb{R}$, such that

$$\|f(x)\|_{\mathbb{R}^d} \leq M_x \|f\|_{\mathbb{H}} \tag{9}$$

for all $f \in \mathbb{H}$. Then, using the Riesz representation theorem, for any direction $v \in \mathbb{R}^d$, there exists a unique function in $\mathbb{H}$ denoted $\phi^*_{x,v}$ such that

$$v^T f(x) = \langle f, \phi^*_{x,v} \rangle \tag{10}$$

Define the matrix-valued kernel $K$ such that

$$K_{ij}(x, x') = \left\langle \phi^*_{x,e_i}, \phi^*_{x',e_j} \right\rangle, i, j = 1 \ldots d \tag{11}$$

where $e_1, \ldots, e_d$ is the natural basis of $\mathbb{R}^d$. Then $K$ is a kernel, $\mathbb{H}$ is the vvRKHS of $K$ and $\phi^*_{x,v}(\cdot) = K(\cdot, x)v$. The reproducing property of the kernel is, for any $f \in \mathbb{H}$, $x \in \mathcal{X}$, and $v \in \mathbb{R}^d$,

$$v^T f(x) = \langle f, K(\cdot, x)v \rangle \tag{12}$$

.

### 2.3 Occupation kernel functions for vvRKHS

The notion of occupation kernel functions, introduced in (Rosenfeld et al., 2019a), is central to this work. Let $\mathbb{H}$ be a vvRKHS of functions $\mathbb{R}^d \to \mathbb{R}^d$ with kernel $K$. Consider a parametric curve $x \colon [a, b] \to \mathbb{R}^d$, $t \mapsto x(t)$ and define the operator $L_x$ from $\mathbb{H}$ to $\mathbb{R}^d$, $L_x(f) = \int_a^b f(x(t))dt$. $L_x$ is linear. If $x \mapsto K_{ii}(x, x)$ is continuous for each $i = 1 \ldots d$, then $L_x$ is also continuous[1]. For each $v \in \mathbb{R}^d$, the occupation kernel function of the curve $x$ in the direction $v$, denoted $L^*_{x,v}$ is then defined as the element in $\mathbb{H}$ such that $v^T L_x(f) = \langle f, L^*_{x,v} \rangle$ for all $f \in \mathbb{H}$. By the Riesz representation theorem, this element exists and is unique.

Let us now provide a useful characterization of $L^*_{x,v}$ in terms of the curve $x$, the vector $v$ and the kernel matrix $K$. Note that for any $w \in \mathbb{R}^d$

$$
\begin{aligned}
w^T L^*_{x,v}(y) &= \langle L^*_{x,v}, K(\cdot, y)w \rangle \\
&= \langle K(\cdot, y)w, L^*_{x,v} \rangle \\
&= v^T L_x(K(\cdot, y)w) \\
&= v^T \int_a^b K(x(t), y)w\,dt \\
&= w^T \left[ \int_a^b K(y, x(t))dt \right] v
\end{aligned}
\tag{13}
$$

thus

$$L^*_{x,v}(\cdot) = \left[ \int_a^b K(\cdot, x(t))dt \right] v \tag{14}$$

Here, we used the reproducing property (12), the symmetry of the inner product, the definition of the occupation kernel and the functional $L_x$, and the symmetry of the kernel $K$. It is convenient to notate the matrix of functions

$$M_x(\cdot) = \int_a^b K(\cdot, x(t))dt \tag{15}$$

so that the $L^*_{x,v}(\cdot) = M_x(\cdot)v$. When $K$ is an I-separable kernel, this reduces to $L^*_{x,v}(\cdot) = \left( \int_a^b k(\cdot, x(t))dt \right) v$

Using the same arguments (see Appendix D.2), we find

$$\langle L^*_{x,v}, L^*_{y,w} \rangle = v^T \left[ \int_a^b \int_a^b K(x(s), y(t))ds\,dt \right] w \tag{16}$$

---

[1]A derivation is provided in Appendix D.1

It is also convenient to use the notation

$$M_{x,y} = \int_a^b \int_a^b K(x(s), y(t)) ds dt \tag{17}$$

so that

$$\langle L_{x,v}^*, L_{y,w}^* \rangle = v^T M_{x,y} w. \tag{18}$$

Note that when $K$ is I-separable,

$$\langle L_{x,v}^*, L_{y,w}^* \rangle = \left( \int_a^b \int_a^b k(x(s), y(t)) ds dt \right) v^T w. \tag{19}$$

## 3 Methods

Consider the ODE

$$\dot{x} = f_0(x), x \in \mathbb{R}^d \tag{20}$$

where $f_0 : \mathbb{R}^d \to \mathbb{R}^d$ is a fixed, unknown vector field. Consider also $n$ curves

$$x_1(t), \ldots, x_n(t), [a_i, b_i] \to \mathbb{R}^d, i \in \{1, \ldots, n\} \tag{21}$$

### 3.1 The multivariate occupation kernel (MOCK) algorithm

We describe the MOCK algorithm in two steps. In the first step, we assume that we observe the curves in (21). We aim to recover the vector field $f_0$ driving the ODE in (20). Assuming that this vector field belongs to a vvRKHS, and under a penalized least square loss, we provide a solution expressed in terms of occupation kernel functions. In the second step, we relax the hypothesis and assume that snapshots along these trajectories are provided in place of the trajectories. Rearranging these trajectories and replacing the integrals with numerical quadratures makes the problem tractable.

### 3.2 The occupation kernel algorithm from curves

Let us design an optimization setting, an inverse problem, for recovering $f_0$ from these trajectories. Let $\mathbb{H}$ be a vvRKHS of continuous vector-valued functions $\mathbb{R}^d \to \mathbb{R}^d$, $\lambda > 0$ a constant, and let us define the functional,

$$J(f) = \frac{1}{n} \sum_{i=1}^n \left\| \int_{a_i}^{b_i} f(x_i(t)) dt - x_i(b_i) + x_i(a_i) \right\|_{\mathbb{R}^d}^2 + \lambda \|f\|_{\mathbb{H}}^2 \tag{22}$$

The first term is minimized when $f = f_0$. This is a consequence of the fundamental theorem of calculus. However, aside from degenerate cases, this minimizer is not unique. Thus, the second term is a regularization term. The problem of minimizing (22) over $\mathbb{H}$ is well-posed. It has a unique solution that can be expressed using the occupation kernel functions as follows:

**Theorem 1.** *The unique minimizer of $J$ over the vvRKHS $\mathbb{H}$ with kernel $K$ is*

$$f^*(\cdot) = \sum_{i=1}^n L_{x_i, \alpha_i}^*(\cdot) = \sum_{i=1}^n M_{x_i}(\cdot) \alpha_i, \alpha_i \in \mathbb{R}^d \tag{23}$$

*where $L_{x_i, \alpha_i}^*$ is the occupation kernel function of the curve $x_i$ along the interval $[a_i, b_i]$ in the direction $\alpha_i$, that is*

$$L_{x_i, \alpha_i}^*(\cdot) = \left[ \int_{a_i}^{b_i} K(\cdot, x_i(t)) dt \right] \alpha_i = M_{x_i}(\cdot) \alpha_i \tag{24}$$

*and where the vector $\alpha = (\alpha_1^T, \ldots, \alpha_n^T)^T$ is the solution of*

$$(M + \lambda n I)\,\alpha = x(b) - x(a),$$
$$x(a) = \left(x_1(a_1)^T, \ldots, x_n(a_n)^T\right)^T,$$
$$x(b) = \left(x_1(b_1)^T, \ldots, x_n(b_n)^T\right)^T \tag{25}$$

*and $M$ is the $(nd, nd)$ matrix made of $(d, d)$ blocks, each defined by*

$$M_{ij} = M_{x_i, x_j} = \int_{t=a_j}^{b_j} \int_{s=a_i}^{b_i} K\left(x_i(s), x_j(t)\right) ds dt \tag{26}$$

To build some intuition about how the ridge penalty term ensures well-posedness of the minimization problem, suppose that many vector fields can generate trajectories perfectly fitting the data, making the first term in the cost function equal to zero. Occam's razor dictates choosing the simplest such vector field. Here, simplicity corresponds to the smallest vvRKHS norm as measured by the second term. The strict convexity of the squared vvRKHS norm ensures that no two solutions can have the same minimal norm. Indeed, any convex combination of such solutions would yield a smaller norm, contradicting the minimality assumption. Therefore, the minimizer is unique.

The proof of Theorem 1 is a natural generalization of the representer theorem in RKHSs. It is presented in Appendix E. Figure 1 illustrates the result of the theorem. In the case of an implicit I-separable kernel, the linear system in (25) decouples into $d$ linear systems, each of size $n \times n$ where $n$ is the number of trajectories. This allows for an algorithm linear in $d$. If the I-separable kernel derives from an explicit scalar-valued kernel, the linear systems are $q \times q$ where $q$ is the number of dimensions of the scalar-valued kernel. The experiments in Section 4 exploit this remarkable situation. The experiments in Section 5 use non-separable kernels. This is a tractable design when $d$ is not too large.

### 3.3 The occupation kernel algorithm from data

Let us now assume that instead of observing $n$ curves that are solutions of (20), we observe $m + 1$ snapshots, sometimes noisy, $z_i = \left(z_i^{(0)}, \ldots, z_i^{(m)}\right)$ at time-points $t_0, \ldots, t_m$ coming from a true trajectory $x_i$, $i = 1 \ldots n$. Firstly, we reshape this data into observations coming from $N = mn$ trajectories, each made of two samples. This is done by viewing every couple of consecutive observations as two observations associated with a single trajectory. In other words, we assume that we observe (possibly with noise) the initial and final points of $N$ trajectories $x_i, i = 1 \ldots N$, that we denote by $z_i = (z_i^{(0)}, z_i^{(1)})$, the times at which $z_i^{(0)}$ and $z_i^{(1)}$ are sampled are denoted by $t_i^{(0)}$ and $t_i^{(1)}$. $z_i$ is a matrix of dimensions $d$ by 2.

Secondly, we replace the integral in (24) by an integral quadrature, noted $\oint$, and the double integral in (26) by a double integral quadrature, noted $\oiint$. Observations $z_i$ are used to compute the quadrature involving the trajectory $x_i$. In both cases, we use the trapezoidal rule, providing

$$\oint K(\cdot, x_i(t)) dt = \frac{(t_i^{(1)} - t_i^{(0)})}{2} \left(K(\cdot, z_i^{(0)}) + K(\cdot, z_i^{(1)})\right) \tag{27}$$

and

$$\oiint K\left(x_i(s), x_j(t)\right) ds dt = \frac{(t_i^{(1)} - t_i^{(0)})(t_j^{(1)} - t_j^{(0)})}{4} \left(K(z_i^{(0)}, z_j^{(0)}) + K(z_i^{(0)}, z_j^{(1)})\right.$$
$$\left. + K(z_i^{(1)}, z_j^{(0)}) + K(z_i^{(1)}, z_j^{(1)})\right) \tag{28}$$

The resulting algorithms for learning the vector field and predicting a trajectory given an initial condition are presented in Algorithm 1 and Algorithm 2 respectively. Algorithm 1 and Algorithm 2 are simplified and do not contain the optimizations implemented to reduce the computational complexity (see Section 3.4). The linear system in line 5: of Algorithm 1 is solved using `numpy.linalg.solve` from NumPy v1.26.4

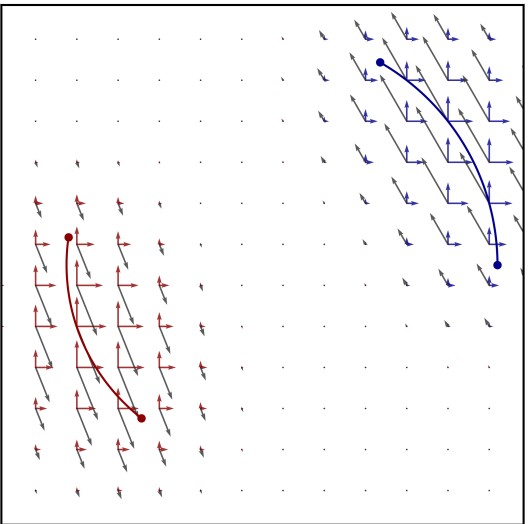

Figure 1: Illustration of the MOCK algorithm with the Gaussian separable kernel. We observe the red and blue trajectories, respectively $x_1$ and $x_2$ (running counterclockwise). The horizontal and vertical red vector fields correspond respectively to the occupation kernel functions $L_{x_1,e_1}^*$ and $L_{x_1,e_2}^*$ and similarly for the horizontal and vertical blue vector fields $x_2$. The influence of each trajectory is local. The grey vector field is the MOCK algorithm solution, a linear combination of the red and blue vector fields.

---

**Algorithm 1** MOCK learning: Estimate the parameters defining the vector field.

---

**Require: Training data**. $z_i, i = 1 \dots N$ ($N$ matrices of dimension $(d, 2)$)

1: Compute $\delta = \left( z_1^{(1)} - z_1^{(0)}, \dots, z_N^{(1)} - z_N^{(0)} \right)^T \in \mathbb{R}^{N \times d}$

2: **for** $i = 1 \dots N, j = 1 \dots N$ **do**

3:      $M_{ij} \leftarrow \iint K\left(x_i(s), x_j(t)\right) ds dt$

4: **end for**

5: Solve the linear system $(M + \lambda N I)\, \alpha = \delta$ for $\alpha$

6: **Return**: $\alpha$

---

**Algorithm 2** MOCK inference: Generate trajectories given initial conditions.

---

**Require: Training data**. $z_i, i = 1 \dots N$. ($N$ matrices of dimension $(d, 2)$)

**Require: Output of Algorithm 1**: $\alpha = (\alpha_1, \dots, \alpha_n)^T$.

**Require: Initial conditions**. $p$ vectors :$(y_1^0, \dots, y_p^0)$

1: **for** $j = 1 \dots p$ **do**

2:      Using a numerical integrator, generate the solution of:

$$\begin{cases} \dot{y}_j = f^*(y_j) \\ y_j(0) = y_j^0 \end{cases}$$

3:      $f^*(y_j) = \sum_{i=1}^N \left[ \oint K(y_j, x_i(t)) dt \right] \alpha_i$

4: **end for**

---

### 3.4 Computational complexity

We specialize to the separable kernels case with $A = I$, see Section 2.2. In such a case, the linear system in (25) becomes equivalent to solving an $N \times N$ linear system where the right-hand side is $N \times d$. Therefore, this requires $O\left(dN^2\right)$ to construct the kernel matrix, $O\left(N^2\right)$ to estimate the integrals, and $O\left(dN^3\right)$ to solve the linear system. Overall, the MOCK algorithm is thus linear in $d$. The space complexity is $O\left(N^2\right)$.

On the other hand, if we let $\psi(x) \in \mathbb{R}^{d \times q}$, then $K(x, y) = \psi(x)^T \psi(y) \in \mathbb{R}^{d \times d}$ is an explicit kernel. In such a case, a $q \times q$ linear system is solved to find $f$, requiring $N$, $d \times q$ by $q \times d$ matrix-matrix multiplications to construct the system. This provides a computational complexity of $O(dNq^2 + q^3)$. Further details are provided in Appendix F.

## 4 Experiments

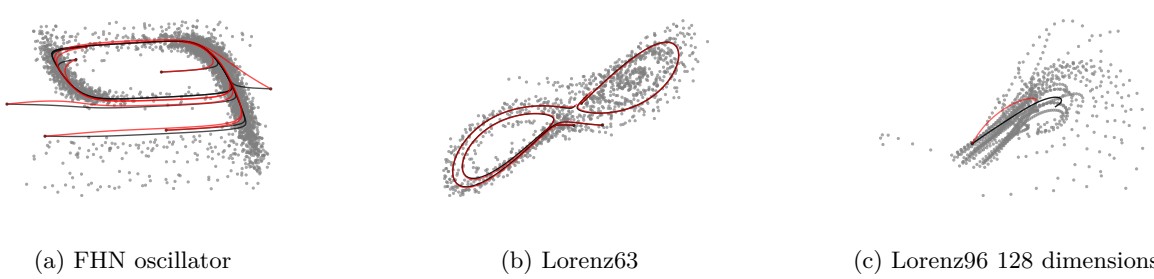

| (a) FHN oscillator | (b) Lorenz63 | (c) Lorenz96 128 dimensions |

Figure 2: The grey points are the training data. The black curves are the test trajectories. The red curves are the predictions on the test set. For data in dimensions higher than two, only the first two dimensions are shown.

### 4.1 Datasets

We test the methods on a diverse set of synthetic and real-world datasets that reflect the challenges of learning systems of ODEs. Table 1 provides the dimension of the state space, the number of trajectories, and the average number of snapshots per trajectory for each of the 9 datasets considered. In each data set, the trajectories were partitioned into training (60%), validation (20%), and testing (20%) sets.

| Name | Type | d | n | m |
|---|---|---|---|---|
| NFHN | Synthetic | 2 | 150 | 201 |
| Lorenz63 | Synthetic | 3 | 150 | 201 |
| Lorenz96-16 | Synthetic | 16 | 100 | 243 |
| Lorenz96-32 | Synthetic | 32 | 100 | 243 |
| Lorenz96-128 | Synthetic | 128 | 100 | 243 |
| Lorenz96-1024 | Synthetic | 1024 | 100 | 100 |
| CMU | Real | 50 | 75 | 106.7 |
| Plasma | Real | 6 | 425 | 2.95 |
| Imaging | Real | 117 | 231 | 2.26 |

Table 1: For each dataset, d is the number of dimensions of the system, which is the dimension of the state-space, n is the total number of trajectories, and m is the average number of samples per trajectory. In each data set, the trajectories were partitioned into training (60%), validation (20%), and testing (20%) sets.

**Noisy FitzHugh-Nagumo (NFHN)** The FitzHugh-Nagumo oscillator (FitzHugh, 1961) is a nonlinear 2D dynamical system that models the basic behavior of excitable cells, such as neurons and cardiac cells[2]. We add considerable noise to this otherwise simple dataset (see Figure 3).

**Noisy Lorenz63 (Lorenz63)** The Lorenz63 system (Lorenz, 1963) is a 3D system provided as a simplified model of atmospheric convection.

**Lorenz96** The Lorenz96 data arises from (Lorenz, 1995) in which a system of equations is proposed that may be chosen to have any dimension greater than 3. (Lorenz96-16) has 16 dimensions, etc.

**CMU Walking (CMU)** The Carnegie Mellon University (CMU) Walking data is a repository of publicly available data. It can be found at http://mocap.cs.cmu.edu/.

**Plasma** This dataset includes imaging and plasma biomarkers from the WRAP study, see (Johnson et al., 2018).

**Imaging** This dataset includes regional imaging biomarkers from the WRAP study, see (Johnson et al., 2018).

### 4.2 Data format

Each dataset is stored as a rectangular matrix. Each row corresponds to a data point. The columns include the id of the trajectory, the time, the features, and three binary columns indicating whether the row is to be used for training, validation, or testing.

### 4.3 Kernels, regularization, and hyperparameter tuning for the MOCK algorithm

We train the MOCK models using the scalar-valued Matérn kernels, including the Gaussian, Laplace, and $C^{10}$ kernels (Fuselier et al., 2016). The analytic expressions are provided in Appendix C.2. We also train an explicit kernel with 200 Fourier Random features (Rahimi & Recht, 2007). We use Bayesian optimization for hyperparameter tuning of the regularization parameter $\lambda$ and bandwidth parameter $\sigma$. $\lambda$ controls the smoothness of the solution, while $\sigma$ controls the relevant scale for the input data. Specifically, we use the **gp_minimize** function from the Scikit-Optimize (**skopt**) package, **version 0.10.2**, which implements Gaussian Process optimization.

### 4.4 Comparable methods

We select competitive methods covering various categories. In addition, we provide a brief ablation study comparing our method to a standard regression approach for fitting the slopefield (see Appendix H).

**Sparse Identification of Nonlinear Dynamics (SINDy)** SINDy is a well-developed class of data-driven methods for identifying dynamical systems models from trajectories. (Brunton et al., 2016) These methods rely on sparse regression techniques to isolate the most relevant terms in the governing equations from a set of candidate functions. SINDy demonstrates robustness for sparse and limited data (Kaheman et al., 2020), (Fasel et al., 2022).

**Reduced Order Models (ROMs)** ROMs are a class of methods for simplifying high-dimensional dynamical systems by projecting them onto a lower-dimensional space. Dynamic mode decomposition (DMD) is a data-driven method for extracting spatiotemporal patterns and coherent structures from high-dimensional data generated by dynamical systems (Tu, 2013). eDMD extends DMD to handle nonlinear dynamics by working in a higher-dimensional feature space. (Williams et al., 2015). We benchmark with eDMD-RFF (Lew et al., 2023), eDMD-Poly (Williams et al., 2015), and eDMD-Deep (Yeung et al., 2019)

**Deep Learning Methods** Deep learning methods can be used in conjunction with ROMs to learn transformations to lower-dimensional spaces. These methods use deep learning to construct an efficient

---

[2]Further details of the all datasets are provided in Appendix B.

representation of the dynamical system and to capture nonlinear dynamics (Lusch et al., 2018), (Yeung et al., 2019), (Li et al., 2019). Deep learning methods can also be incorporated into the SINDy framework to identify sparse, interpretable, and predictive models from data Champion et al. (2019), (Bakarji et al., 2022). We benchmark with ResNet (He et al., 2016) (Lu et al., 2021)

**Latent ODEs for Irregularly-Sampled Time Series** The Latent ODE method (also a deep learning method (Rubanova et al., 2019)) is an update of the Neural ODEs model introduced in (Chen et al., 2018). The core idea is to represent the hidden dynamics of time series data in a continuous latent space using ODEs. Given the latent trajectory, the observations of the time series are assumed to be independent. The latent trajectory dymanics are governed by a neural ODE model. The model uses an encoder-decoder framework where the encoder maps observed data to a latent initial value via an RNN architecture. The hidden states are then carried through neural ODEs between times of observations. Finally, the decoder generates the latent trajectory forward in time and predicts future observations.[3]

The hyperparameters for each method are shown in Table 4 in Appendix G. All experiments are run in Google Colab using the default settings. The GPU was only enabled (A100, High-RAM) for the deep learning techniques. Lastly, to get an idea of the difficulty of each problem, we compare all methods against the null model, which predicts no change, that is $x(t) = x_0$ for all $t \in [a, b]$. Datasets for which no models are able to do much better than this null model are difficult datasets to learn and perhaps less useful for benchmarking learning methods.

## 4.5 Evaluation Method

For each dataset, the trajectories were partitioned into training, validation, and testing. At test time, we integrated the predicted trajectories starting at the given initial conditions and compared to the true trajectories. To measure the performance of each method, we define two types of errors for each trajectory of the test set. We define:

$$\text{Err} = \sqrt{\sum_{i=2}^{m} (t_i - t_{i-1}) \|y_i - \hat{y}_i\|^2} \tag{29}$$

where $y_i$ and $\hat{y}_i$ are the observed and predicted trajectory samples at time $t_i$, respectively. Additionally, we define 1-Err by setting $m = 2$ in (29). All the methods were trained, validated, and tested on the same data. Table 2 provides the average Err and 1-Err errors across the trajectories of the test set. To test whether the MOCK methods are (statistically) significantly better than comparable methods and vice versa, we first generated a Wilcoxon test p-value for every MOCK method (four different kernels) against the other comparable methods. The pairs of observations used in the test are the errors of the two compared methods for every trajectory in the test set. Finally, to generate a p-value comparing all MOCK methods together against another comparable method, we used Fisher's method (Mosteller & Fisher, 1948) to combine the four p-values of the MOCK methods. We report a $*$ in Table 2 if the group of MOCK methods is significantly better ($p < 0.01$) and $\dagger$ if the compared method outperforms MOCK with statistical significance ($p < 0.01$). Otherwise, no method is significantly better than the other.

## 4.6 Evaluation Performance

Examples of the output of the MOCK algorithm are presented visually in Figures 2 and 4. Table 2 summarizes the prediction errors for each experiment. While no single method outperforms other methods over all datasets, MOCK outperforms all other methods in 3 of the 9 datasets (on the task of full trajectory prediction, denoted Err), and outperforms all other methods on 4 of the 9 datasets on the task of predicting the next sample (denoted 1-Err). It also performs competitively on the remaining datasets on both tasks. Notably, eDMD-Deep yielded the best results for the CMU experiments. Deep learning models and DMD methods learn the dynamics in a feature space and therefore have less stringent constraints on the dynamics. We believe this is why they outperform with datasets such as the CMU dataset as our model assumes the

---

[3]Latent ODEs was implemented using the Github repository: https://github.com/YuliaRubanova/latent_ode. The subsampling was disabled for the Plasma and Imaging datasets because the training trajectories were too short in these datasets.

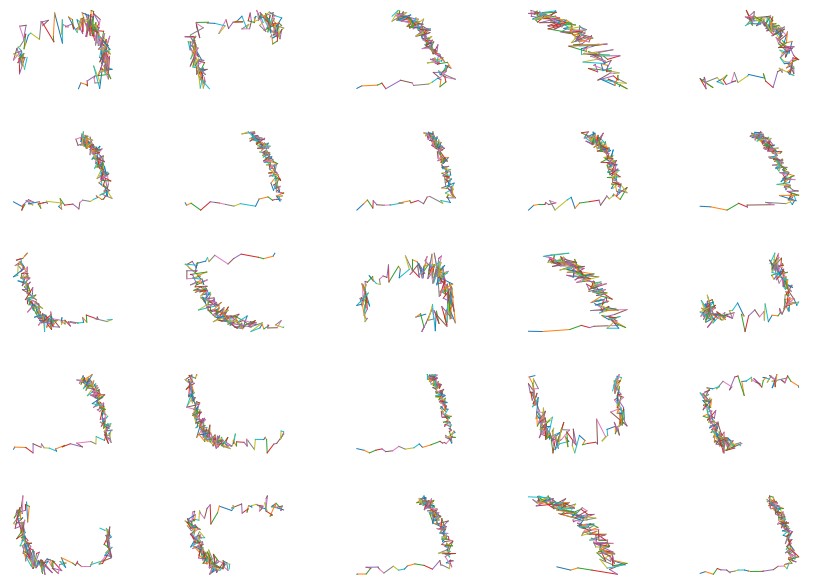

(a) 25 randomly sampled trajectories in the NFHN (Noisy FHN) training set.

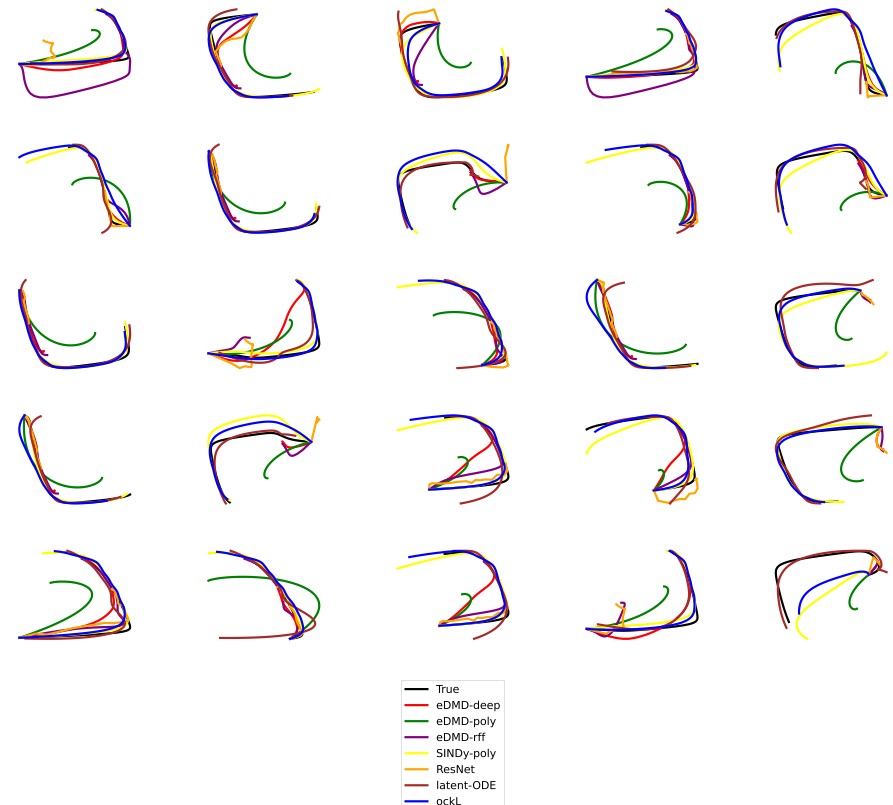

(b) 25 randomly sampled trajectories in the test set for NFHN. Black: true trajectory, red: eDMD-deep, green: eDMD-poly, purple: eDMD-rff, yellow: SINDy-poly, orange: ResNet, brown:latent-ODE, and blue:MOCK-L.

Figure 3: NFHN dataset

Table 2: Dynamical System Estimation Results

**Lorenz63**

| | Err | 1-Err |
|---|---|---|
| ResNet | 2.07* | .014* |
| eDMD-Deep | 2.91* | .009* |
| eDMD-Poly | 2.22* | .012* |
| eDMD-RFF | 1.93* | .011* |
| null | 2.56* | .011* |
| SINDy-Poly | .99* | **.002** |
| MOCK-G | .66 | .002 |
| MOCK-L | **.65** | .002 |
| MOCK-M | 2.52 | .012 |
| MOCK-F | 2.52 | .012 |
| Lode | 1.49* | .092* |

**NFHN**

| | Err | 1-Err |
|---|---|---|
| ResNet | 5.94* | .064* |
| eDMD-Deep | 4.54* | .036* |
| eDMD-Poly | 4.10* | .037* |
| eDMD-RFF | 4.01* | .037* |
| null | 9.69* | .043* |
| SINDy-Poly | 1.68* | **.022**† |
| MOCK-G | 1.40 | .029 |
| MOCK-L | 1.11 | .030 |
| MOCK-M | 1.41 | .029 |
| MOCK-F | 1.38 | .028 |
| Lode | **.64**† | .131* |

**Lorenz96-16**

| | Err | 1-Err |
|---|---|---|
| ResNet | 6.42* | .010* |
| eDMD-Deep | 5.93* | .040* |
| eDMD-Poly | 6.91* | .036* |
| eDMD-RFF | 6.22* | .029* |
| null | 7.50* | .039* |
| SINDy-Poly | .40* | .0003* |
| MOCK-G | .69 | .0007 |
| MOCK-L | **.22** | **.0001** |
| MOCK-M | .22 | .0001 |
| MOCK-F | .22 | .0001 |
| Lode | 3.82* | .138* |

**Lorenz96-32**

| | Err | 1-Err |
|---|---|---|
| ResNet | 11.56* | .020* |
| eDMD-Deep | 8.79* | .057* |
| eDMD-Poly | 9.67* | .051* |
| eDMD-RFF | 8.10* | .028* |
| null | 10.49* | .056* |
| SINDy-Poly | 8.56* | .037* |
| MOCK-G | .47 | **.0001** |
| MOCK-L | .48 | .0001 |
| MOCK-M | .49 | .0001 |
| MOCK-F | **.47** | .0001 |
| Lode | 6.06* | .217* |

**Lorenz96-128**

| | Err | 1-Err |
|---|---|---|
| ResNet | 24* | 2.64* |
| eDMD-Deep | 17* | .089* |
| eDMD-Poly | 19* | .109* |
| eDMD-RFF | 16† | .201* |
| null | 21* | .112* |
| SINDy-Poly | 19* | .066* |
| MOCK-G | 17 | .064 |
| MOCK-L | 16 | **.035** |
| MOCK-M | 16 | .040 |
| MOCK-F | 16 | .036 |
| Lode | **15**† | .342* |

**Lorenz96 -1024**

| | Err | 1-Err |
|---|---|---|
| ResNet | 66.1* | 6.48* |
| eDMD-Deep | 26.9* | .41* |
| eDMD-Poly | 61.5* | .42* |
| eDMD-RFF | 54.2* | .38* |
| null | 67.0* | .16* |
| SINDy-Poly** | NA | NA |
| MOCK-G | 18.0 | **.013** |
| MOCK-L | 17.8 | .014 |
| MOCK-M | 17.9 | .013 |
| MOCK-F | 18.0 | .015 |
| Lode | **15.8**† | 1.04* |

**Plasma**

| | Err | 1-Err |
|---|---|---|
| ResNet | 4.93* | 2.89* |
| eDMD-Deep | 4.09 | 2.42 |
| eDMD-Poly | 4.01 | **2.39** |
| eDMD-RFF | 4.07 | 2.43 |
| null | 4.00* | 2.41 |
| SINDy-Poly | **3.95** | 2.40 |
| MOCK-G | 3.96 | 2.41 |
| MOCK-L | 3.96 | 2.41 |
| MOCK-M | 3.96 | 2.41 |
| MOCK-F | 3.96 | 2.41 |
| Lode | 4.28* | 2.68* |

**Imaging**

| | Err | 1-Err |
|---|---|---|
| ResNet | 27.4* | 19.4* |
| eDMD-Deep | 15.6* | 12.5 |
| eDMD-Poly | 15.7 | 12.5 |
| eDMD-RFF | 27.9* | 19.3* |
| null | 16.2* | 12.6* |
| SINDy-Poly | 96.1* | 53.7* |
| MOCK-G | 15.6 | 12.3 |
| MOCK-L | 15.6 | 12.3 |
| MOCK-M | 15.7 | 12.3 |
| MOCK-F | 15.8 | 12.4 |
| Lode | **14.8**† | **11.7** |

**CMU**

| | Err | 1-Err |
|---|---|---|
| ResNet | 22.6* | .86† |
| eDMD-Deep | 15.7† | 2.03* |
| eDMD-Poly | **14.9**† | .78† |
| eDMD-RFF | 14.9† | **.76**† |
| null | 21.6* | 1.01* |
| SINDy-Poly | 33.8* | .91* |
| MOCK-G | 21 | .89 |
| MOCK-L | 19.6 | .93 |
| MOCK-M | 21.5 | 1.01 |
| MOCK-F | 21.6 | 1.01 |
| Lode | 14.9† | 1.88* |

**Description:** Minimum values are in bold. A $*$ indicates a significant (Fisher's method) p-value ($< 0.01$) in favor of the MOCK methods in the comparison. A † indicates a significant p-value in favor of the method compared with all MOCK methods. There is no statistically significant difference otherwise. Our models are labelled as MOCK-G - occupation kernel method with Gaussian kernel, MOCK-M - occupation kernel method with Matérn kernel, MOCK-F - occupation kernel method with Random Fourier Features (RFF), and MOCK-L - occupation kernel with Laplace kernel (See section 4.2). We compare against SINDy-Poly, eDMD-Deep, eDMD-Poly, eDMD-RFF, ResNet and Latent Ode (Lode) (See section 4.3). $**$ No result could be obtained for SINDy-Poly for this dataset due to computational complexity issues.

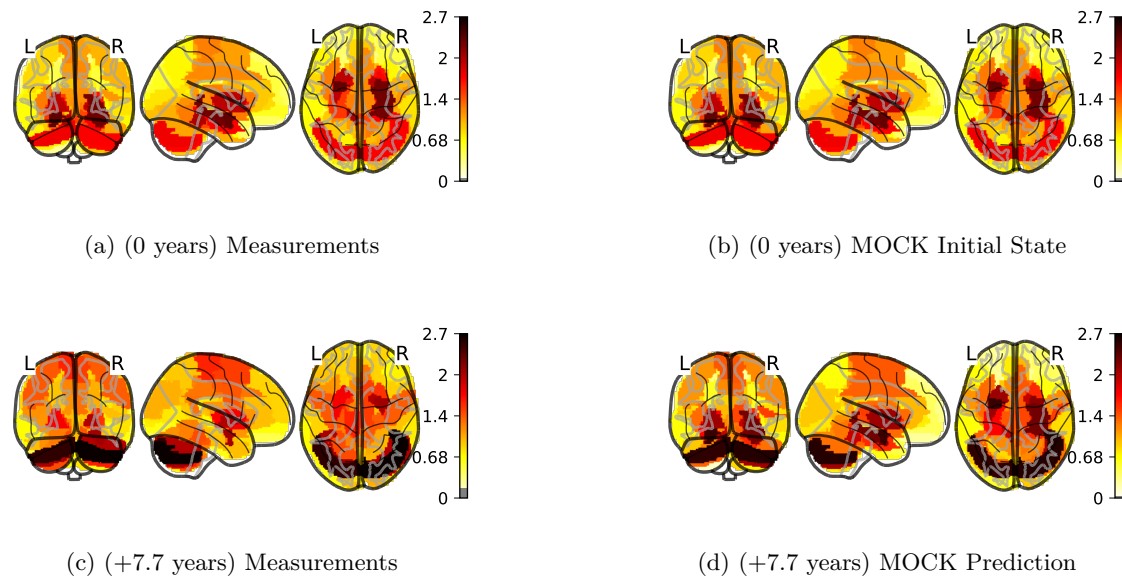

(a) (0 years) Measurements

(b) (0 years) MOCK Initial State

(c) (+7.7 years) Measurements

(d) (+7.7 years) MOCK Prediction

Figure 4: MOCK Predicted Biomarker Values from the imaging dataset.

dynamics arises from a homogeneous system, and this is not the case with the CMU dataset. See Appendix B.2.

The differences observed in Table 2 are not always significant. Recall that we use the sign † when a method is better than all the MOCK methods. We count 5 models out of 9 with at least one † for Err and only 2 out of 9 for the 1-Err. Note that the MOCK algorithm is optimized for the 1-Err since the trajectories of the training set are systematically reshaped into trajectories of two observations. Thus it is not surprising that it performs better for this metric.

When comparing the kernels used for the MOCK method, the Laplace kernel performs the best overall, consistent with other analyses of this kernel (Geifman et al., 2020). Note also that the random Fourier features kernel, which approximates the Gaussian kernel, performs well for all datasets, allowing for linear implementations in $N$, which is the number of examples in the training set. Moreover, we use the same number of features (200) in all cases, leading to models with parameter sizes of $200 \cdot d$ for Fourier random feature implementations. Importantly, because our model size scales linearly with d we do not need to increase the number of Fourier random features for higher dimensional problems.

The trajectories predicted by MOCK-L for NFHN (in blue in Figure 3b) are visually close to the ground truth (in black) in all but the last case. SINDy-poly (in yellow) is competitive. ResNet provides very irregular trajectories. Latent ODE provides good long-term predictions. However, the initial points do not coincide with the initial points of the ground truth provided for all the algorithms. In other words, while Latent-ODE performs very well on the task of predicting full trajectories, it performs very poorly on the next sample prediction task which is an inherent limitation of some deep learning models like Latent-ODE and Resnet.

We tested MOCK with a dataset of 1024 dimensions to show how well it scales. The implementation of SINDy-Poly that we used did not provide a result for this data due to runtime issues; see NA in the Table 2.

### 4.7 Comparative computational complexity

The computational complexities are compared in Table 3. The MOCK algorithm with implicit and explicit kernel is linear in $d$, the dimension of the state space. Resnet and Lode are quadratic in $d$. In principle, SINDy-Poly, eDMD-Poly, eDMD-RFF, and eDMD-Deep are linear in $d$. However, $q$, the number of parameters in the learned model, needs to increase with $d$ to obtain acceptable performances. This means the complexity is, in practice, more than linear in $d$. In the case of SINDy-Poly with quadratic basis functions, $q$ is $\mathcal{O}(d^2)$.

| MOCK implicit | MOCK explicit | ResNet | eDMD-Deep |
|:---:|:---:|:---:|:---:|
| $\mathcal{O}(dN^3)$ | $\mathcal{O}(dNq^2 + dq^3)$ | $\mathcal{O}\left(kd^2TN\right)$ | $\mathcal{O}(TN(Ldq + q^2) + dq^2 + q^3)$ |
| **eDMD-Poly** | **eDMD-RFF** | **Lode** | **SINDy-Poly** |
| $\mathcal{O}((N + d)q^2 + N^2q + q^3)$ | $\mathcal{O}((N + d)q^2 + q^3)$ | $\mathcal{O}\left(nT\left(k_1d^2 + k_2q^2\right)\right)$ | $\mathcal{O}(T(qd + q^3 + dNq^2))$ |

Table 3: Runtimes of training for each method, excluding hyper-parameters validation. $d$ is the dimension of the state-space. $N$ is the total number of samples in all the trajectories. $q$ is the number of features. $T$ is the number of epochs. $k_1$ and $k_2$ are constant.

## 5 Learning vector fields with constraints

Divergence-free ($\nabla \cdot v = 0$) and curl-free ($\nabla \times v = 0$) vector fields appear in a diverse set of applications including fluid dynamics (Wendland, 2009), (Fuselier et al., 2016), magnetohydrodynamics (McNally, 2011) and modeling magnetic fields (Wahlström et al., 2013), image processing (Polthier & Preuß, 2003), and surface reconstruction (Drake et al., 2022). Accordingly, a significant amount of research has been devoted to the development of curl-free and divergence-free kernel methods (Narcowich & Ward, 1994), (Lowitzsch, 2005; Narcowich et al., 2007), (Fuselier, 2008), (Fuselier & Wright, 2009), (Gao et al., 2022), (Scheuerer & Schlather, 2012), as well as to the development of Gaussian processes constrained by PDEs (Harkonen et al., 2023), (Henderson et al., 2023) in recent years. We will demonstrate how the occupation kernel method can be adapted to ensure that the recovered vector field analytically satisfies the divergence-free/curl-free constraint for an appropriate choice of matrix kernel $K$. Then, for ease of presentation, we will apply just the divergence-free kernels to both real and synthetic datasets.

### 5.1 Divergence-free kernels

Let $x \in \mathbb{R}^d$ and $d = 2, 3$, and let $\phi(\|x\|)$ be a radial basis function. Then the standard construction (Fuselier, 2008) for divergence-free and curl-free matrix-valued radial basis functions are

$$\{-\Delta I + \nabla\nabla^T\}\phi(\|x\|) \quad \text{and} \quad -\nabla\nabla^T\phi(\|x\|), \tag{30}$$

respectively, where $\Delta = \sum_{i=1}^d \partial^2/\partial x_i^2$ and $[\nabla\nabla^T]_{ij} = \partial^2/\partial x_i \partial x_j$, $1 \le i, j \le d$. For example, if $d = 2$, then

$$\{-\Delta I + \nabla\nabla^T\}\phi(\|x\|) = \begin{bmatrix} -\frac{\partial^2 \phi}{\partial x_2^2} & \frac{\partial^2 \phi}{\partial x_1 \partial x_2} \\ \frac{\partial^2 \phi}{\partial x_2 \partial x_1} & -\frac{\partial^2 \phi}{\partial x_1^2} \end{bmatrix}(\|x\|). \tag{31}$$

Thus, the columns of the matrix-valued radial basis function $\{-\Delta I + \nabla\nabla^T\}\phi(\|x\|)$ are divergence-free. Similarly, we have that the columns of $-\nabla\nabla^T\phi(\|x\|)$ are curl-free.

We select the divergence-free kernel $K(x, y) = \{-\Delta I + \nabla\nabla^T\}\phi(\|x - y\|)$ where our choice of scalar radial basis function is the $C^{10}$ Matérn kernel (Fuselier et al., 2016).

### 5.2 Hamiltonian systems

A system

$$\dot{x}_1 = f(x_1, x_2), \quad \dot{x}_2 = g(x_1, x_2) \tag{32}$$

is called a *Hamiltonian system* if there exists a function $H(x_1, x_2)$ (called the *Hamiltonian*) for which $f = \partial H/\partial x_2$ and $g = -\partial H/\partial x_1$. This implies that the vector field $(f, g)$ is divergence-free. Furthermore, along every orbit we have $H(x_1, x_2) =$ constant and any conservative dynamical equation $\ddot{u} = f(u)$ leads to a Hamiltonian system where the Hamiltonian coincides with the total energy. For example, every orbit of the conservative system corresponding to the equation describing the motion of a pendulum

$$\dot{x}_1 = x_2, \quad \dot{x}_2 = -\frac{g}{\ell}\sin(x_1) \tag{33}$$

satisfies the conservation law

$$H(x_1, x_2) = \frac{1}{2}x_2^2 - \frac{g}{\ell}\cos(x_1) = E \tag{34}$$

for some constant $E$.

### 5.3 Experiments

We test the divergence-free MOCK method on the following real and synthetic datasets.

**Pendulum Problem** We generate data using (33) with $\ell = 1$ and $g = 9.8$. This 2-dimensional synthetic dataset consists of 100 trajectories of 50 samples each. This dataset allows us to test whether the MOCK method can, simultaneously, learn a known Hamiltonian system while analytically enforcing the divergence free constraint.

**Buoy Data** We obtained a buoy dataset from https://oceanofthings.darpa.mil/data#tab-all, which contains two-dimensional trajectories of buoys submerged in the ocean. We sampled every tenth observation of the raw data and converted the time measurements into years. Trajectories with only one sample were dropped.

### 5.4 Evaluation

The error was computed using (29) for both datasets. For the pendulum problem, the scalar Matérn kernel provided an error of $1.5 \times 10^{-1}$ while the divergence-free kernel provided an error of $1.9 \times 10^{-2}$, an improvement by an order of magnitude. Not only does the divergence-free kernel allow for a physically constrained solution, but it is also more accurate. For the buoy data, the scalar Matérn kernel provided an error of $3.5 \times 10^1$ while the divergence-free kernel provided an error of $3.1 \times 10^1$. Thus, despite the presence of noise (which is not necessarily divergence-free), we still achieve a reduction in the error of about 10%.

Figure 5 shows the error in the computed vector fields of the pendulum problem. A plot of the results for the buoy data problem using the divergence-free kernel, along with the training data, is shown in Figure 6.

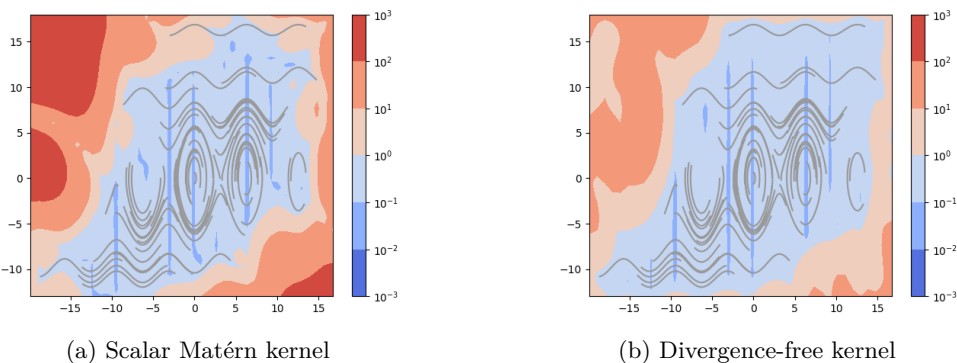

(a) Scalar Matérn kernel        (b) Divergence-free kernel

Figure 5: The magnitude of the error in vector field approximation of the pendulum problem obtained using different choices of kernel along with the training set (gray).

## 6 Summary and discussion

The implicit matrix-valued kernel formulation has enabled us to demonstrate that the MOCK algorithm, originally proposed by J. Rosenfeld and collaborators, effectively learns the vector field of an ODE in multivariate settings or under predefined constraints, while maintaining linear scalability with the dimension of the state space. We have benchmarked the algorithm against a representative selection of competitive methods and provided compelling experimental evidence showcasing its superior performance on many datasets, along with consistently competitive results across all cases tested.

A rigorous analysis of the algorithm's convergence and error, including precise assumptions regarding noise, is deferred to future work. This analysis will incorporate the quadrature error and the effects of noise, as

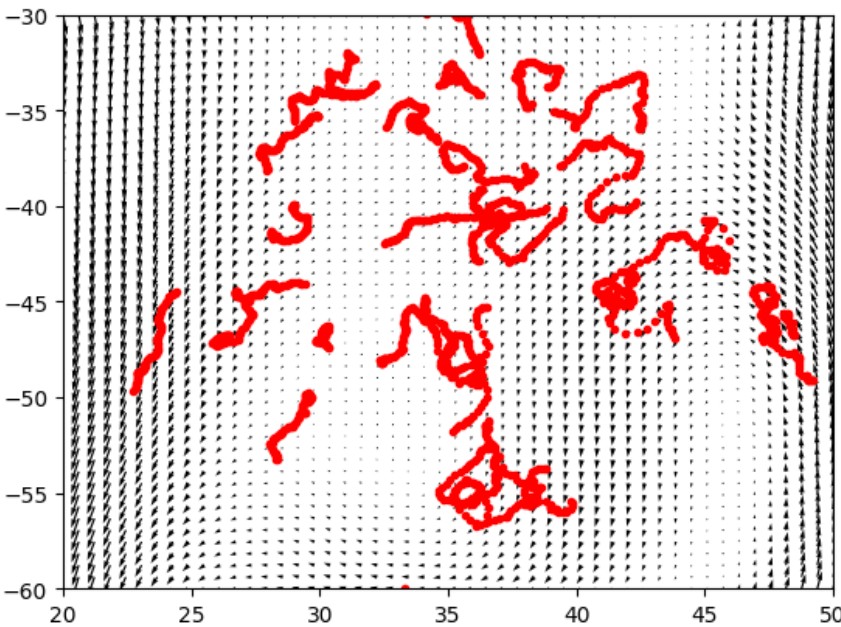

Figure 6: A detail of the vector field learned using the divergence-free kernel from the buoy data together with the training set (red).

well as provide concentration inequalities to quantify the error between predicted and true trajectories as functions of quadrature precision, observation noise levels, and data granularity, offering insights into the algorithm's generalization capabilities.

The surprising simplicity of the MOCK technique, combined with its strong performance, opens the door to numerous opportunities for optimization, generalization, and further exploration. Future work is expected to advance the state-of-the-art in high-dimensional dynamics learning, extend applications to PDEs, and analyze generalization properties in detail.

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

## A    Experiment Code

We created a code capsule on Code Ocean that hosts all the methods used in our experiments for system identification of ordinary differential equations. The code repository allows for the repeatability of our results for a single experiment—Lorenz63—and facilitates the evaluation of the performance of different models on synthetic and real-world datasets. The real world plasma and imaging experiment are restricted data and will not be available for the evaluation.

We provide an *anonymized* zipfile (6.6 MB) export of the code capsule for review at `https://drive.google.com/file/d/1p36HH00dHGLuBeHEfhJyMlfxjv1At5La/view?usp=sharing`. The code can be run locally in a Docker container with instructions in the README.md and REPRODUCING.md.

To ensure the reproducibility of the experiment, we have tuned the hyperparameters for each method, except for ResNet, which is known to require substantial tuning time. The code repository provides detailed instructions for running each method experiment, including the necessary input data and the corresponding hyperparameter settings. Please note that due to limitations on Code Ocean, we have set the TensorFlow library to use the CPU rather than the GPU, as using the GPU resulted in numerous errors. This may increase the runtime of the experiments. For the paper, we ran some experiments in google colab on A100 GPUs where the training time was much lower.

Upon running the code repository, the experiments will take approximately 1.5 hours to complete. The results of each experiment are provided in the form of trajectory CSV files, capturing the predicted trajectories of the identified systems. These trajectory files can be further analyzed or visualized as desired. Additionally, a summary of the performance of each method is available in the code repository.

## B  Datasets

### B.1  Synthetic data

**Noisy FitzHugh-Nagumo**   The FitzHugh-Nagumo oscillator (FitzHugh, 1961) is a nonlinear 2D dynamical system that models the basic behavior of excitable cells, such as neurons and cardiac cells (See Figure 3). The system is popular in the analysis of dynamical systems for its rich behavior, including relaxation oscillations and bifurcations. The FHN oscillator consists of two coupled ODEs that describe the membrane potential $v$ and recovery variable $w$,

$$\dot{v} = v - \frac{v^3}{3} - w + RI \tag{35}$$

$$\tau\dot{w} = v + a - bw \tag{36}$$

Here, $I$ is the external current input, $a$ and $b$ are positive constants that affect the shape and duration of the action potential, and $\tau$ is the time constant that determines the speed of the recovery variable's response. We added random Gaussian noise of standard deviation 0.12 to this dataset.

**Noisy Lorenz63**   The 3D Lorenz 63 system (Lorenz, 1963) is a simplified model of atmospheric convection, but has since become a canonical example of chaos in dynamical systems. The Lorenz 63 system exhibits sensitive dependence on initial conditions and parameters, which gives rise to its characteristic butterfly-shaped chaotic attractor. The equations are

$$\begin{aligned} \frac{\mathrm{d}x}{\mathrm{d}t} &= \sigma(y - x) \\ \frac{\mathrm{d}y}{\mathrm{d}t} &= x(\rho - z) - y. \\ \frac{\mathrm{d}z}{\mathrm{d}t} &= xy - \beta z \end{aligned} \tag{37}$$

Here, $x$, $y$, and $z$ are state variables representing the fluid's convective intensity, and $\sigma$, $\rho$, and $\beta$ are positive parameters representing the Prandtl number, the Rayleigh number, and a geometric factor, respectively. We added random Gaussian noise with standard deviation 0.5 to this dataset.

**Lorenz96**   The Lorenz96 data arises from (Lorenz, 1995) in which a system of equations is proposed that may be chosen to have any dimension greater than 3. The chaotic system is defined by:

$$\frac{dx_k}{dt} = -x_{k-1}x_{k-2} + x_{k+1}x_{k-1} - x_k + F \tag{38}$$

where we take $F = 8$ and consider 16, 32, 128 and 1024 dimensional systems. Indices are assumed to wrap so that for an $n$ dimensional system $x_{-2} = x_{n-2}$.

### B.2  Real Data

**CMU Walking**   The Carnegie Mellon University (CMU) Walking data is a repository of publicly available data. It can be found at http://mocap.cs.cmu.edu/. It was generated by placing sensors on a number of subjects and recording the position of the sensors as the subjects walked forward by a fixed amount and then walked back. There were 50 spatial dimensions of data as recorded by the sensors, which were recorded at regular times. Since the observation times were not provided, we separated each observation in time by 0.1 units and began each trajectory at time zero. The CMU Walking data which we used for our experiments consists of 75 trajectories with an average of 106.7 observations per trajectory. We could not use the full amount of data provided because some subjects did not walk in the same manner as other subjects.

It should be noted that it may not be appropriate to consider the CMU walking data as being generated by a single dynamical system. Since there were multiple subjects in the dataset, it is reasonable to conclude that there are multiple dynamical systems responsible for the motion of these different subjects. However, we fit our model to this dataset assuming that it was generated by a single dynamical system.

**Plasma** This dataset includes imaging and plasma biomarkers from the WRAP study, see (Johnson et al., 2018). The imaging biomarker is a distribution volume ratio obtained from Pittsburgh Compound-B positron emission tomography. The plasma biomarkers include $A\beta_{40}/A\beta_{42}$ that reflects specific amyloid beta ($A\beta$) proteoforms; ptau217, which has a high correlation with amyloid PET positivity, GFAP, which measures the levels of the astrocytic intermediate filament glial fibrillary acidic protein, and NFL, a recognized biomarker of subcortical large-caliber axonal degeneration, for a total of 6 biomarkers. There are a total $n = 425$ trajectories, one per subject, and, on average, 2.95 time points per trajectory. The data was split $(70\%, 10\%, 20\%)$ corresponding resp. to training, validation, and testing.

**Imaging** This dataset includes regional imaging biomarkers from the WRAP study, see (Johnson et al., 2018). The imaging biomarker is an atlas-based distribution volume ratio obtained from Pittsburgh Compound-B positron emission tomography. There are a total of 117 biomarkers. There are a total of $n = 231$ trajectories, one per subject, and, on average, 2.26 time points per trajectory. The data was split $(70\%, 10\%, 20\%)$ corresponding resp. to training, validation, and testing.

## C Reproducing Kernel Hilbert Spaces (RKHS)

Basic notions and notations associated with RKHSs are important for understanding the algorithms presented in this paper. We thus provide a short presentation. We limit ourselves to RKHSs over the field of real numbers. RKHSs are Hilbert spaces for which the evaluation functional is continuous. The evaluation functional at $x \in \mathbb{R}$ is a mapping from an RKHS $\mathbb{H}$ to $\mathbb{R}$, which associates to a function its evaluation at $x$, that is $f \mapsto f(x)$. Riesz representation theorem allows us to interpret evaluations of a function in an RKHS as a geometric operation consisting of computing an inner product. That is, there is a unique vector $k_x \in \mathbb{H}$ such that $f(x) = \langle f, k_x \rangle$. Moreover, let us define, for any $x, y \in \mathbb{R}$, the so-called kernel $k(x, y) = \langle k_x, k_y \rangle$. Let us use this to characterize the function $k_x$. Evaluating $k_x$ at $y$ and using Riesz representation provides $k_x(y) = \langle k_x, k_y \rangle = \langle k_y, k_x \rangle = k(y, x)$. Thus the function $k_x(\cdot)$ is the function $k(\cdot, x)$. Moreover, for any $f \in \mathbb{H}$, $f(x) = \langle f, k(\cdot, x) \rangle$. This is the reproducing property of the kernel. Applying this property to $k_y$ implies that $k(x, y) = \langle k(\cdot, x), k(\cdot, y) \rangle$

### C.1 Linear functionals and occupation kernels

Let $\mathbb{H}$ be an RKHS of functions from $\mathbb{R}$ to $\mathbb{R}$ with kernel $k$. Assume that $x \mapsto k(x, x)$ is continuous. Consider a continuous parametric curve $[0, 1] \to \mathbb{R}$, $t \mapsto x(t)$ and define the functional from an RKHS $\mathbb{H}$ to $\mathbb{R}$, $L_x(f) = \int_0^1 f(x(t))dt$. $L_x$ is clearly linear. The Cauchy-Schwarz inequality implies $L_x$ is bounded. Indeed,

$$|L_x(f)| = \left| \int_0^T f(x(t))dt \right| \tag{39}$$

$$= \left| \int_0^T \langle f, k(\cdot, x(t)) \rangle \, dt \right| \tag{40}$$

$$\leq \int_0^T |\langle f, k(\cdot, x(t)) \rangle| \, dt \tag{41}$$

$$\leq \int_0^T ||f|| ||k(\cdot, x(t))|| dt \tag{42}$$

$$= ||f|| \int_0^T \sqrt{k(x(t), x(t))} dt \tag{43}$$

where we have used the Cauchy-Schwartz inequality in (42). Now, since $t \mapsto x(t)$ and $x \mapsto k(x, x)$ are continuous, the integral in (43) is upper-bounded. Thus the functional $L_x(f)$ is continuous.

We use Riesz representation theorem to define the occupation kernel function as the unique element $L_x^*$ in $\mathbb{H}$ that verifies $L_x(f) = \langle f, L_x^* \rangle$. Note that

$$L_x^*(y) = \langle L_x^*, k(\cdot, y) \rangle = \langle k(\cdot, y), L_x^* \rangle = L_x(k(\cdot, y)) = \int_0^1 k(x(t), y)dt \tag{44}$$

Furthermore,

$$\langle L_x^*, L_y^* \rangle = L_y(L_x^*) = \int_0^1 L_x^*(y(t))dt = \int_0^1 \int_0^1 k(x(s), y(t))dsdt \tag{45}$$

## C.2   Kernels

For all experiments (except for Section 5) we use standard scalar-valued positive definite kernels, references to which may be found, see, for example, Table 3.1 on page 42 of (Fasshauer & McCourt, 2015). Additionally, in section 5, we demonstrate how a divergence-free kernel allows us to incorporate physical constraints in our model. The specific divergence-free kernel we use is given in (Fuselier et al., 2016). For the sake of clarity, we include the analytical forms of the kernels used throughout the paper below:

Guassian kernel:

$$g(x, y, \sigma) = exp\left(-\frac{\|x - y\|^2}{2\sigma^2}\right) \tag{46}$$

Laplace kernel:

$$l(x, y, \gamma) = exp\left(-\frac{\|x - y\|}{\gamma}\right) \tag{47}$$

Fourier Feature kernel:

$$f(x, y, \sigma) = \frac{1}{\sqrt{q}} \left[cos(z_1^T x/\sigma + \beta_1), ..., cos(z_q^T x/\sigma + \beta_q)\right]^T \left[cos(z_1^T y/\sigma + \beta_1), ..., cos(z_q^T y/\sigma + \beta_q)\right] \tag{48}$$

Matérn kernel $C^{10}$:

$$m(x, y, \sigma) = \frac{1}{945}exp(-r/\sigma)\left(\left(\frac{r}{\sigma}\right)^5 + 15\left(\frac{r}{\sigma}\right)^4 + 105\left(\frac{r}{\sigma}\right)^3 + 420\left(\frac{r}{\sigma}\right)^2 + 945\left(\frac{r}{\sigma}\right) + 945\right) \tag{49}$$

where in (48) $z_i \sim N(0, I)$ and $\beta_i \sim U(0, 2\pi)$, and in (49) $r = \|x - y\|$. We now introduce the divergence-free kernel of Section 5. Let

$$d_1(x, y, \sigma) = \frac{-1}{945\sigma^2}exp(-r/\sigma)\left(\left(\frac{r}{\sigma}\right)^4 + 10\left(\frac{r}{\sigma}\right)^3 + 45\left(\frac{r}{\sigma}\right)^2 + 105\left(\frac{r}{\sigma}\right) + 105\right)$$

and

$$d_2(x, y, \sigma) = \frac{1}{945\sigma^4}exp(-r/\sigma)\left(\left(\frac{r}{\sigma}\right)^3 + 6\left(\frac{r}{\sigma}\right)^2 + 15\left(\frac{r}{\sigma}\right)^2 + 15\right),$$

and define

$$\phi_{11}(x, y, \sigma) = -d_2(x, y, \sigma)(x_2 - y_2)^2 - d_1(x, y, \sigma),$$
$$\phi_{22}(x, y, \sigma) = -d_2(x, y, \sigma)(x_1 - y_1)^2 - d_1(x, y, \sigma),$$

and

$$\phi_{12}(x, y, \sigma) = \phi_{21}(x, y, \sigma) = d_2(x, y, \sigma)(x_1 - y_1)(x_2 - y_2),$$

where $x = [x_1, x_2]^T$, $y = [y_1, y_2]^T$, and $r = \|x - y\|$. Then,

$$m_{div}(x, y, \sigma) = \begin{bmatrix} \phi_{11} & \phi_{12} \\ \phi_{21} & \phi_{22} \end{bmatrix}(x, y, \sigma). \tag{50}$$

# D  Vector-Valued Reproducing Kernel Hilbert Spaces

There are three ways to characterize a vvRKHS.

Firstly, we can construct a vvRKHS with linear functions of the kernel:

*Definition* 3. Let

$$\mathbb{H}_0 = \left\{ f; f(x) = \sum_{i=1}^{n} K(x, x_i) w_i, x_i \in \mathcal{X}, w_i \in \mathbb{R}^d, i = 1 : n \right\} \tag{51}$$

then, consider the encoding in $\mathbb{H}_0$ of the functions $f$ and $g$,

$$f \longleftrightarrow \left\{ \begin{array}{l} x_1, \ldots, x_n \\ w_1, \ldots, w_n \end{array} \right. \text{ and } g \longleftrightarrow \left\{ \begin{array}{l} y_1, \ldots, y_m \\ v_1, \ldots, v_m \end{array} \right. \tag{52}$$

Define the inner product

$$\langle f, g \rangle = \sum_{i=1}^{n} \sum_{j=1}^{m} w_i^T K(x_i, y_j) v_j \tag{53}$$

then the closure of $\mathbb{H}_0$ for $\langle ., . \rangle$ is the vvRKHS of K.

The second construction starts from a Hilbert space and uses the Riesz representation theorem.
*Definition* 4. Let $\mathbb{H}$ be a Hilbert space of functions $\mathbb{R}^d \to \mathbb{R}^d$ such that for any $x \in \mathbb{R}^d$, the evaluation functional $f \mapsto f(x)$ is continuous: there is a constant $M_x \in \mathbb{R}$, such that

$$||f(x)||_{\mathbb{R}^d} \le M_x ||f||_{\mathbb{H}} \tag{54}$$

for all $f \in \mathbb{H}$. Then, using the Riesz representation, for any $v \in \mathbb{R}^d$ there exists a unique $K_{x,v} \in \mathbb{H}$ such that

$$v^T f(x) = \langle f, K_{x,v} \rangle_{\mathbb{H}} \tag{55}$$

This equation is the *reproducing property.*

Define the matrix-valued kernel

$$K_{ij}(x, x') = \langle K_{x,e_i}, K_{x',e_j} \rangle, i, j = 1 \ldots d \tag{56}$$

where $e_1, \ldots, e_d$ is the natural basis of $\mathbb{R}^d$. Then $K$ is a kernel and $\mathbb{H}$ is the vvRKHS of $K$.

The third formulation allows for checking that a Hilbert space is a vvRKHS.
*Definition* 5. Let $\mathbb{H}$ be a Hilbert space of functions $\mathbb{R}^d \to \mathbb{R}^d$. Let $K$ be a $(d, d)$ matrix-valued kernel. $\mathbb{H}$ is the vvRKHS of $K$ when

1. For any $x' \in \mathcal{X}$, $v \in \mathbb{R}^d$, $x \mapsto K(x, x')v \in \mathbb{H}$

2. The reproducing property holds: for any $f \in \mathbb{H}$, $x \in \mathcal{X}$, $v \in \mathbb{R}^d$, (55) holds

## D.1  Verifying the continuity for the occupation kernel in multiple dimensions

Let $(e_1, \ldots, e_d)^T$ be the standard basis of $\mathbb{R}^d$. Then

$$|e_j^T L_x(f)| = \left| \int_0^T e_j^T f(x(t)) dt \right| \tag{57}$$

$$\le \int_0^T \left| e_j^T f(x(t)) \right| dt \tag{58}$$

$$= \int_0^T |\langle f, K(., x(t)) e_j \rangle| dt \tag{59}$$

$$\le ||f|| \int_0^T \sqrt{K_{jj}(x(t), x(t))} dt \tag{60}$$

where we have used the Cauchy-Schwartz inequality. If $x \mapsto K_{jj}(x, x)$ is continuous, then since $t \mapsto x(t)$ is continuous, $e_j^T L_x$ is bounded for each $1 \leq j \leq d$, which implies that $L_x$ is bounded and thus continuous.

### D.2 Occupation Kernel inner product

Consider

$$L_{x,v}^*(\cdot) = \int_0^T K(\cdot, x(t))dtv$$

$$L_{y,w}^*(\cdot) = \int_0^T K(\cdot, y(t))dtw$$

we wish to evaluate

$$\langle L_{x,v}^*, L_{y,w}^* \rangle = w^T L_y \left( L_{x,v}^*(\cdot) \right) \tag{61}$$

$$= w^T \int_0^T \left\{ \int_0^T K(y(s), x(t))dtv \right\} ds \tag{62}$$

$$= w^T \left( \int_0^T \int_0^T K(y(s), x(t))dtds \right) v \tag{63}$$

## E Proof of Theorem 1

Consider the linear span of the occupation kernel functions $L_{x_i, \alpha_i}^*, i = 1 \ldots n$

$$\mathcal{F} = \left\{ f \in \mathbb{H}, f = \sum_{i=1}^n L_{x_i, \alpha_i}^*, \alpha = (\alpha_1, \ldots, \alpha_n) \in \mathbb{R}^{d \times n} \right\} \tag{64}$$

Note that $\mathcal{F}$ is linear and finite-dimensional; thus, it is a closed linear subspace of $\mathbb{H}$. We can then project any function in $\mathbb{H}$ orthogonally onto it

$$f = f_{\mathcal{F}} + f_{\mathcal{F}^\perp} \tag{65}$$

Now, for each $i = 1 \ldots n$, and $v \in \mathbb{R}^d$,

$$v^T \int_{a_i}^{b_i} f(x_i(t))dt = v^T L_{x_i}(f) = \langle L_{x_i,v}^*, f \rangle = \langle L_{x_i,v}^*, f_{\mathcal{F}} + f_{\mathcal{F}^\perp} \rangle = \langle L_{x_i,v}^*, f_{\mathcal{F}} \rangle \tag{66}$$

where the previous to last equality comes from the fact that $f_{\mathcal{F}^\perp}$ is perpendicular to $L_{x_i,v}^*$. Thus,

$$\int_{a_i}^{b_i} f(x_i(t))dt = L_{x_i}(f_{\mathcal{F}}) \tag{67}$$

Next, using the Pythagorean equality

$$||f||^2 = ||f_{\mathcal{F}}||^2 + ||f_{\mathcal{F}^\perp}||^2 \geq ||f_{\mathcal{F}}||^2 \tag{68}$$

Thus, for any $f \in \mathbb{H}$, $J(f_{\mathcal{F}}) \leq J(f)$ which proves that the minimum of $J$ actually belongs to $\mathcal{F}$. Moreover, note that

$$||f_{\mathcal{F}}||^2 = \left\langle \sum_{i=1}^n L_{x_i, \alpha_i}^*, \sum_{i=1}^n L_{x_i, \alpha_i}^* \right\rangle = \sum_{i,j=1}^n \left\langle L_{x_i, \alpha_i}^*, L_{x_j, \alpha_j}^* \right\rangle$$

$$= \sum_{i,j=1}^n \alpha_i^T \left[ \int_{a_i}^{b_i} \int_{a_j}^{b_j} K(x_i(s), x_j(t))dsdt \right] \alpha_j = \sum_{i,j=1}^n \alpha_i^T M_{ij} \alpha_j = \alpha^T M \alpha \tag{69}$$

Also,

$$L_{x_i}(f_{\mathcal{F}}) \quad = \quad L_{x_i}\left(\sum_{j=1}^{n} L^*_{x_j,\alpha_j}\right) \tag{70}$$

$$= \quad \int_{a_i}^{b_i} \sum_{j=1}^{n} L^*_{x_j,\alpha_j}(x_i(t))dt \tag{71}$$

$$= \quad \sum_{j=1}^{n}\left[\int_{a_i}^{b_i}\int_{a_j}^{b_j} K(x_i(t),x_j(s))dsdt\right]\alpha_j \tag{72}$$

$$= \quad \sum_{j=1}^{n} M_{ij}\alpha_j \tag{73}$$

$$= \quad [M\alpha]_i \tag{74}$$

The minimization problem in $f$ is thus equivalent to minimizing in $\alpha$

$$J(\alpha) = \frac{1}{n}\sum_{i=1}^{n}\|[M\alpha]_i - x_i(b_i) + x_i(a_i)\|^2_{\mathbb{R}^d} + \lambda\alpha^T M\alpha \tag{75}$$

or equivalently,

$$J(\alpha) = \frac{1}{n}\|M\alpha - x(b) + x(a)\|^2_{\mathbb{R}^{nd}} + \lambda\alpha^T M\alpha \tag{76}$$

since $\lambda > 0$ and $M$ is PSD, this is solved by

$$(M + \lambda nI)\,\alpha = x(b) - x(a) \tag{77}$$

## F  Computational Complexity

We detail below an analysis of the complexity for MOCK in terms of $d$, the dimension of the state space, and $N$, the total number of observations. We present the implicit and explicit kernels successively.

### F.1  Implicit kernel

In several of our experiments, an implicit Gaussian kernel was used. The Gram matrix for the kernel is $N \times m$ where $N$ is the total number of samples in the dataset $X$ and $m$ is the total number of samples in the dataset $Y$. If $X \in \mathbb{R}^{d \times N}$, and $Y \in \mathbb{R}^{d \times m}$, we may compute any radial basis function kernel defined by:

$$G_{i,j} = f(\|x_i - y_j\|^2) \tag{78}$$

by observing:

$$D = \text{outer}(\text{sum}(X \circledast X), 1_m) - 2X^T Y + \text{outer}(1_N, \text{sum}(Y \circledast Y)) \tag{79}$$

Where $D_{i,j} = \|X_i - Y_j\|^2$ is the matrix of square distances, $\circledast$ is the Hadamard product, $sum$ is column sum, $outer$ is an outer product, and $1_N$ is the ones vector in $N$ dimensions. $G \in \mathbb{R}^{N \times N}$ is then calculated by applying $f$ component by component to $D$ for the training set $X$ with $X$. The computational bottleneck of this Gram matrix computation is the $N \times d$ by $d \times m$ matrix matrix multiplication, with a worst case run time that is $O(dNm) = O(dN^2)$ when $Y = X$ (with a naive implementation of matrix matrix multiplication).

We integrate over intervals of a single pair of samples, and these integrals are estimated using the trapezoid rule quadrature. Therefore, for instance, if a single trajectory is given, we would get our estimate of all our integrals by simply taking the matrix

$$k = \frac{h^2}{4}\begin{bmatrix} 1 & 1 \\ 1 & 1 \end{bmatrix} \tag{80}$$

and convolving it with the Gram matrix $G$. If multiple trajectories are given, we apply the same convolution to $G$ and ignore terms involving sums that mix samples from different trajectories. In NumPy indexing notation, we may apply the above convolution with the expression:

$$M = \frac{h^2}{4} \left( G[1:,1:] + G[:,-1,1:] + G[1:,:-1] + G[:-1,:-1] \right) \tag{81}$$

This adds an additional $O(N^2)$ computational complexity (for fixed $G \in \mathbb{R}^{N \times N}$).

Finally, the linear system we solve is:

$$(M + \lambda I) A = Y \tag{82}$$

Where $M \in \mathbb{R}^{N \times N}$ and $A, Y \in \mathbb{R}^{N \times d}$, which adds the dominating computational complexity term of $O(dN^3)$.

### F.2 Explicit Kernel

Assuming the vvRKHS has a matrix-valued kernel of the form

$$K(x,y) = \psi(x)\psi^T(y) \tag{83}$$

where

$$\psi : \mathbb{R}^d \to \mathbb{R}^{d \times q} \tag{84}$$

we may recall:

$$f(x) = \sum_i \int_{a_i}^{b_i} \psi(x)\psi(x_i(\tau))^T d\tau \alpha_i = \psi(x) \sum_i \int_{a_i}^{b_i} \psi(x_i(\tau))^T d\tau \alpha_i = \psi(x)\beta \tag{85}$$

where $\beta \in \mathbb{R}^q$ is defined by $\beta = \sum_i \int_{a_i}^{b_i} \psi(x_i(\tau))^T d\tau \alpha_i$. Letting

$$\Psi \in \mathbb{R}^{Nd \times q} \tag{86}$$

be defined by:

$$\Psi_i \in \mathbb{R}^{d \times q} = \int_{a_i}^{b_i} \psi(x_i(\tau)) d\tau \tag{87}$$

Let

$$L^* = \Psi \Psi^T \tag{88}$$

then

$$\alpha^T L^* \alpha = \beta^T \beta. \tag{89}$$

Thus, we reduce the minimization problem (22) to

$$\min_\beta J(\beta) = \frac{1}{N} \sum_{i=1}^N \|[\Psi \beta]_i - x_i(b_i) + x_i(a_i)\|_{\mathbb{R}^d}^2 + \lambda \beta^T \beta \tag{90}$$

This simplifies to

$$\left( \Psi^T \Psi + \lambda N I \right) \beta = \Psi^T y \tag{91}$$

where $y \in \mathbb{R}^{Nd}$ with $y_i \in \mathbb{R}^d$ and $y_i = x_i(b_i) - x_i(a_i)$. Thus, in the explicit kernel case, we require evaluating $\Psi$ which is $O(Ndq)$, a $q \times Nd$ by $Nd \times q$ matrix-matrix multiplication $O(q^2(Nd))$ and a $q \times q$ linear system solve which is $O(q^3)$. Notice, in the explicit kernel case, no assumptions on the overall structure of the kernel matrix $K(x,y)$ are needed. For instance, the matrices no longer need to be diagonal or proportional to the identity matrix. However, in practice, for best results, $q$ will typically be dependent on the dimension $d$. If the kernel is I-separable however, the runtime may be improved. Indeed, supposing $\psi(x) = \phi(x) \otimes I \in \mathbb{R}^{pd}$ where $\phi : \mathbb{R}^d \to \mathbb{R}^p$. The runtime reduces from $O(q^2 Nd + q^3) = O((pd)^2 Nd + (pd)^3)$ to $O(p^2 nd + p^3 d)$

| Method | Hyperparameters |
|---|---|
| Null | N/A |
| **Sparse Identification of Nonlinear Dynamics** | |
| SINDy (Brunton et al., 2016) | polynomial degree, sparsity threshold |
| **Reduced Order Models** | |
| eDMD-RFF (DeGennaro & Urban, 2019) | number of features, lengthscale |
| eDMD-Poly (Williams et al., 2015) | polynomial degree |
| **Deep Learning** | |
| ResNet (He et al., 2016) (Lu et al., 2021) | network depth/breadth |
| eDMD-Deep (Yeung et al., 2019) | latent space dimension, autoencoder depth/breadth |
| Latent Ode (Rubanova et al., 2019) | latent space dimension, autoencoder and decoder depth/breadth |

Table 4: Hyperparameters for the comparators

## G   Comparators:

The methods we benchmark against are presented in the Table 4. References and hyperparameters are specified. The Null model is used as a baseline to determine that something was learned of the dataset. Our null model has no parameters and is the trivial dynamical system for which the slope-field is zero everywhere. A model that fails to outperform the null model likely did not learn any useful information about the dynamical system. We present now the computational complexity for each method.

### G.1   SINDy

The computational complexity for the SINDy algorithm is based on the Sequential Threshold Least Squares (STLS) process, and it is influenced by two main factors:

1. Number of library functions ($q$) — This refers to the number of candidate terms in the library, which is formed based on the polynomial degree and dimensionality of the system. In our case, we used polynomials at most degree 3 and we used 16, 32, and 128 dimensions of the Lorenz system.

2. Number of iterations ($T$) — The number of times the STLS algorithm iterates to threshold the coefficients. Each iteration performs regression on a progressively smaller set of terms after thresholding.

The complexity is derived by:

1. Sequential Thresholding: In each iteration, the algorithm identifies and zeroes out small coefficients (based on the threshold), which has complexity O(qd), where q is the number of terms and d is the number of variables (or dimensions).

2. Least Squares Regression: After thresholding, the algorithm re-solves the least squares problem for the remaining large terms. The complexity of this operation depends on the size of the remaining set p, and it scales as O($p^3$), where $p \leq q$. In addition, as with our technique constructing the linear system is an $O(dNp^2)$ operation.

   Hence, after each iteration, the algorithm refines the set of non-zero coefficients in the library, progressively reducing the number of terms included in the least squares regression.

Iterations

   The number of iterations, $T$, can vary depending on the data and thresholding behavior. In general, $T$ is relatively small compared to the size of the library, but it still affects the total complexity.

Overall complexity:

Given that the STLS algorithm iterates $T$ times, the total computational complexity is $\mathcal{O}(T \cdot (qd + q^3 + dNq^2))$. The cubic and quadratic terms $\mathrm{O}(q^3 + dNq^2)$ dominate when solving the least squares problem, especially when the number of terms $q$ is large, which is typical when higher-degree polynomials or large systems are considered.

## G.2 Resnet

The Resnet algorithm use a residual network that learns the vector field. The input is a vector of dimension $d$. The output is also a vector of dimension $d$. There are $q$ fully connected layers. The Resnet is trained during T epochs. The computational complexity is $\mathcal{O}(TNqd^2)$.

## G.3 Latent ODEs

The Latent ODE model consists of two primary components: an encoder and a decoder. In the encoder, a set of ODEs is solved backward in time, where the ODEs operate on the observed state space of dimension $d$. In the decoder, the ODEs are solved forward in time, but they operate on a latent space of dimension $q$.

To compute the overall complexity of the algorithm, we define several key variables:

1. $T$: the number of epochs used to train the neural network,

2. $k_1$: the total number of function evaluations (time points) used in the numerical solver for the encoder ODEs,

3. $k_2$: the number of function evaluations for the decoder ODE,

4. $n$: the number of observed trajectories.

Given these notations, the per-epoch time complexity of the encoder is $\mathcal{O}(nk_1d^2)$, as the ODEs in the encoder are evaluated on a state space of dimension $d$. The per-epoch complexity of the decoder is $\mathcal{O}(nk_2q^2)$, as the ODEs in the decoder are evaluated on a latent space of dimension $q$.

Thus, the total per-epoch complexity of the model is:

$$\mathcal{O}\left(n\left(k_1d^2 + k_2q^2\right)\right)$$

Finally, considering that the training runs for $T$ epochs, the overall complexity of the algorithm becomes:

$$\mathcal{O}\left(nT\left(k_1d^2 + k_2q^2\right)\right)$$

## G.4 eDMD-Poly and eDMD-RFF

The complexity of the eDMD method can be expressed in terms of $N$: the number of training examples, $q$: the number of library functions, and $d$: the dimension of the data. We used a polynomial library for eDMD-Poly and random Fourier features for eDMD-RFF. The underlying algorithm was the same.

The first step is to evaluate the library functions on the training data, obtaining an $(N, q)$ matrix $\Psi$. Next, we must numerically approximate the time derivatives of these quantities, yielding a matrix $\Psi'$. The complexity of these operations is $\mathcal{O}(Nq)$.

In the next step, we compute an approximation of the Koopman operator, given by $\mathcal{K} = \Psi^+\Psi'$. The pseudoinverse has a complexity which depends on whether $N < q$ or $N \geq q$. In the former, the complexity is $\mathcal{O}(N^2q)$, while in the latter, it is $\mathcal{O}(Nq^2)$. The matrix-vector product $\Psi^+\Psi'$ costs $\mathcal{O}(Nq^2)$ to compute.

Then we compute an eigendecomposition of $\mathcal{K}$, costing $\mathcal{O}(q^3)$. The eigenvectors correspond to approximate eigenfunctions of the true Koopman operator. Let $E$ be the matrix of eigenvectors.

Let $B$ be the matrix of coefficients of the full-state observable in terms of the library functions. This is a $(d, q)$ matrix. The eigenmodes are thus given by $BE^{-1}$. The matrix $E^{-1}$ can be expressed as the left eigenvectors of $\mathcal{K}$; thus, it is computed during the eigndecomposition. Therefore, the eigenmodes only require multiplying $(d, q)$ and $(q, q)$ matrices together, costing $\mathcal{O}(dq^2)$.

The matrix $B$, if not computed analytically, may be computed by inverting a $(q, q)$ matrix and performing $d$ matrix-vector products, resulting in a complexity of $\mathcal{O}(dq^2 + q^3)$.

Thus, the total complexity is given by $\mathcal{O}((N + d)q^2 + N^2q + q^3)$ if $N < q$ and $\mathcal{O}((N + d)q^2 + q^3)$ if $N \geq q$.

The pseudoinverse step may be impractical in situations where there are many training examples. Thus, it may be substituted with a low-rank approximation using SVD. The rank of this approximation is a hyperparameter and will affect the complexity of the algorithm. We omit this consideration for the sake of simplicity.

### G.5 eDMD-Deep

The eDMD-Deep method attempts to learn the library functions and the Koopman operator simultaneously. It does so by defining the library functions as the output of a neural network and minimizing the error incurred from a single Koopman operator update. Thus, the complexity is dependent on the architecture of the neural network used and the choice of optimizer.

Assume that optimization is done with stochastic gradient descent and the neural network outputs $q$ library functions. Assume also that the data consists of $n$ observations of $d$-dimensional trajectories. Suppose the neural network has $L$ layers with $q$ neurons each. Then a single iteration of SGD for the neural network has a cost of $\mathcal{O}(Ldq)$. Thus, an epoch as a cost of $\mathcal{O}(NLdq)$. Training for $T$ epochs then costs $\mathcal{O}(TLNdq)$.

One must simultaneously optimize the cost function for the Koopman operator $K$, which is a $(q, q)$-dimensional matrix, with the neural network. Computing the gradient of the cost with respect to $K$ requires multiplying a $(q, q)$ matrix and $q$-dimensional vector, costing $\mathcal{O}(q^2)$. We compute this product for each item in the training set, costing $\mathcal{O}(Nq^2)$. Thus, the total cost of optimizing $K$ over $T$ epochs is $\mathcal{O}(TNq^2)$. Therefore, the cost of training the Koopman operator together with the library functions is $\mathcal{O}(TN(Ldq + q^2))$.

Once one has obtained the library functions and the matrix $K$, one proceeds as in eDMD and computes an eigendecomposition of $K$, followed by computing the eigenmodes, costing $\mathcal{O}(q^3 + dq^2)$. Thus, the complexity of the entirety of the eDMD-Deep algortihm is given by $\mathcal{O}(TN(Ldq + q^2) + dq^2 + q^3)$.

### G.6 Runtime Results for the models

The theoretical runtimes of each benchmarked method is provided in Figure 7. The experimental runtimes are also provided. The training and validation procedures differed for each of the methods making runtime comparisons difficult to interpret. Moreover, the hardware on which the models were run differed as well. For instance, all deep models were trained and run on GPUs while all other methods were not. In Figure 7 we show the experimental runtimes for Lorenz96-16, Lorenz96-32 and Lorenz96-128 to give some suggestion as to how the runtimes scale as the dimension of the problem increases. The runtime of MOCK is not greatly impacted by the increase in the dimension.

The runtime for ResNet and Lode is quadratic in d (as can be seen in Figure 7). In addition, eDMD-Poly and SINDy-Poly suffer from the problem that the number of polynomial features scales at least linearly with d. Thus these models scale at least cubically in d. q is determined by the user for eDMD-RFF and eDMD-Deep. If the user fixes q, then these methods will be linear in d. See Figure 7 and the discussion of Section G.6.

## H Regression Ablation Study:

One simple way to learn a vector field for a dynamical system is to solve a regression problem for the slopes of the trajectories. This regression problem introduces the same regularization to the same RKHSs; however,

| MOCK $\mathcal{O}(dN^3)$ | | |
|---|---|---|
| Data | val (h) | non-val (s) |
| L96-16 | 2.27 | 9.0 |
| L96-32 | 2.5 | 9.1 |
| L96-128 | 3.7 | 9.5 |

| ResNet $\mathcal{O}\left(kd^2TN\right)$ | | |
|---|---|---|
| Data | val (h) | non-val (s) |
| L96-16 | .02 | 86.1 |
| L96-32 | .03 | 107.8 |
| L96-128 | .07 | 232.5 |

| eDMD-Deep $\mathcal{O}(TN(Ldq + q^2) + dq^2 + q^3)$ | | |
|---|---|---|
| Data | val (h) | non-val (s) |
| L96-16 | 5.63 | 62.34 |
| L96-32 | 4.58 | 81.3 |
| L96-128 | 6.8 | 123.9 |

| eDMD-Poly $\mathcal{O}((N + d)q^2 + N^2q + q^3)$ | | |
|---|---|---|
| Data | val (h) | non-val (s) |
| L96-16 | .31 | 7.0 |
| L96-32 | .43 | 8.1 |
| L96-128 | .5 | 9.4 |

| eDMD-RFF $\mathcal{O}((N + d)q^2 + q^3)$ | | |
|---|---|---|
| Data | val (h) | non-val (s) |
| L96-16 | .08 | 1.56 |
| L96-32 | .08 | 1.6 |
| L96-128 | .15 | 2.58 |

| Lode $\mathcal{O}\left(nT\left(k_1d^2 + k_2q^2\right)\right)$ | | |
|---|---|---|
| Data | val (h) | non-val (s) |
| L96-16 | .23 | 829 |
| L96-32 | .44 | 1568 |
| L96-128 | 4.7 | 16891 |

| SINDy-Poly $\mathcal{O}(T(qd + q^3 + dNq^2))$ | | |
|---|---|---|
| Data | val (h) | non-val (s) |
| L96-16 | .1 | 1.4 |
| L96-32 | .0025 | 1.5 |
| L96-128 | .03 | 23 |

Figure 7: val(h) is the training time including the training of the hyperparameters, in hours. non-val(s) is the training time excluding the training of the hyper-paramters, in seconds. For non-val, The MOCK algorithm scales most favorably with the dimension of the dataset, as increasing the dimension from 16 to 128 increases training time by less than 10%.

solves a different loss. We have performed an ablation study, replacing the functional in (22) by

$$J(f) = \frac{1}{n} \sum_{i=1}^{n} \|f(x_i(t)) - \dot{x}_i(t)\|_{\mathbb{R}^d}^2 + \lambda \|f\|_H^2 \tag{92}$$

and estimating $\dot{x}_i(t)$ using finite differences. Specifically, considering only trajectories of two data points, as in Section 3.3, this becomes,

$$J(f) = \frac{1}{n - 1} \sum_{i=1}^{n} \left\| f(z_i^{(0)}) - \frac{z_i^{(1)} - z_i^{(0)}}{t_i^{(1)} - t_i^{(0)}} \right\|_{\mathbb{R}^d}^2 + \lambda \|f\|_H^2 \tag{93}$$

We choose to run the experiment on the noisy FHN data, denoted NFHN; see Section 4.1. We used the Gaussian kernel and the same validation procedure as for the MOCK algorithm. We obtain an error equal to 1.9, where the error is defined in (29). This is to be contrasted with the error for the MOCK algorithms presented in Table 2, which was 1.40 for the same kernel. This experimentally shows the advantages of using the MOCK algorithm compared to regression. Note that this result is not surprising. The use of finite difference for estimating a derivative is known to be problematic for noisy data. Estimating an integral is preferred, and this is what the MOCK algorithm does.

