# OpenReview forum: "MOCK: an Algorithm for Learning Nonparametric Differential Equations via Multivariate Occupation Kernel Functions"
_TMLR — Accepted by TMLR_

### Review · Reviewer_wyKJ · 2025-03-14

**Summary Of Contributions:**

The paper proposes learning kernelised ODE vector field from observed trajectories using the occupation kernel. The authors derive a version of the representer theorem, and present a practical training algorithm.

**Audience:**

Yes

**Broader Impact Concerns:**

No issues.

**Claims And Evidence:**

Yes

**Requested Changes:**

- Authors should discuss what kind of inductive bias the occupation kernel gives to the learned vector field, and how does it differ from a vanilla kernel regression (without the occupation). This would also be useful to show empirically as an ablation or comparison.

- The figure 1 is not trivial to interpret. The L’s seem to go in a different direction from the learned vector field. It also seems that the L fields have constant angle: all the red and blue arrows point in similar direction with just the magnitude changing. What does this mean? Is this a desired effect? Contrasting fig1 with standard kernel regression (without occ) would be useful.

- Authors should discuss the real-world applicability of the method: when and why should a practitioner choose this method? Authors should also discuss the limitations of the method.

- The buoy experiment needs to be revised or justified better. Fig 6 shows strong stochasticity in the data, and I'm not convinced that modelling this as div-free or Hamiltonian makes sense. I'm not sure what kind of conclusion the authors are after here.

**Strengths And Weaknesses:**

S: The method is principled and well-derived contribution to ODE learning

S: The results are competive in a large array of experiments and comparisons. The divergence-free experiment is a useful addition.

W: The method's real-world applicability is not really demonstrated. The paper should discuss in which usecases a practitioners should choose the method, and why

---

> ### Author Response · Authors · 2025-04-17
> **Ablation study and answer to the remaining comments.**
>
> {\em Authors should discuss what kind of inductive bias the occupation kernel gives to the learned vector field, and how does it differ from a vanilla kernel regression (without the occupation). This would also be useful to show empirically as an ablation or comparison.}
>
> The inductive bias is the same as for kernel regression. This bias is disctated by the choice of the kernel, its parameters if any, and the strength of the regularization parameter, denoted $\lambda$ in this paper.
>
> Following the suggestion of the reviewer, we have performed an ablation study, replacing the functional in (21) by
>
> $$    J(f) = \frac{1}{n}\sum_{i=1}^n ||f(x_i(t)) - \dot x_i(t)||_{\mathbb{R}^d}^2 + \lambda ||f||_H^2 $$
>
> and estimating $\dot x_i(t)$ using finite differences. Specifically, considering only trajectories of two data points, as in section 3.3, this becomes,
> $$
>  J(f) = \frac{1}{n-1}\sum_{i=1}^n ||f(z_i^{(0)}) - \frac{z_i^{(1)}-z_i^{(0)}}{t_i^{(1)}-t_i^{(0)}}||_{\mathbb{R}^d}^2 + \lambda ||f||_H^2
> $$
> We choose to run the experiment on the noisy FHN data, denoted NFHN; see section 4.1. We used the Gaussian kernel and the same validation procedure as for the MOCK algorithm. We obtained a RMSE of $Err=1.9$, where $Err$ is defined in (28). This is to be contrasted with the RMSE for the MOCK algorithms presented in Table 2, which is $Err=1.40$ for the same kernel. This experimentally shows the advantages of using the MOCK algorithm compared to regression. Note that this result is not surprising. The use of finite difference for estimating a derivative is known to be problematic for noisy data. Estimating an integral is preferred, and this is what the MOCK algorithm does.
>
>
> {\em The figure 1 is not trivial to interpret. The L’s seem to go in a different direction from the learned vector field. It also seems that the L fields have constant angle: all the red and blue arrows point in similar direction with just the magnitude changing. What does this mean? Is this a desired effect? Contrasting fig1 with standard kernel regression (without occ) would be useful.}
>
> Note first that doing plain regression on the data, here the red and blue curves, would fail to provide a vector field defined on the whole domain.  Next, we agree that getting a good grasp of the notion of occupation kernel functions takes some effort. Occupation kernel functions are vector fields. Hence, they are defined everywhere in the spatial domain. In Figure 1, this domain is a square. Four such vector fields are shown in red (horizontal and vertical) and blue (horizontal and vertical). The estimated vector field, shown in grey, is a linear combination of these four vector fields. In the case of an I-separable kernel, the horizontal and vertical have the same magnitude.
>
> {\em Authors should discuss the real-world applicability of the method: when and why should a practitioner choose this method? Authors should also discuss the limitations of the method.}
>
> Learning differential equations from data has applications in many scientific fields where data is available but the equations are unknown or partially unknown. The introduction of [1] provides many examples.
> The MOCK has broad applicability, similar to the SINDy algorithm described in [1] and the other comparators. To understand which situations MOCK should be preferred over the comparators, we have analyzed performances in Table 2 and computational complexity in Table 3. The MOCK algorithm applies to a wide range of datasets. It performs well with noisy data and generalizes well to high-dimensional and large datasets when using an explicit kernel. It is flexible and can accommodate constraints, including divergence-free vector fields.  However, in this case, the complexity increases as the cube of the dimension.
>
>
> [1] Steven L Brunton, Joshua L Proctor, and J Nathan Kutz. Discovering governing equations from data by
> sparse identification of nonlinear dynamical systems. Proceedings of the national academy of sciences, 113
> (15):3932–3937, 2016.
>
> {\em The buoy experiment needs to be revised or justified better. Fig 6 shows strong stochasticity in the data, and I'm not convinced that modelling this as div-free or Hamiltonian makes sense. I'm not sure what kind of conclusion the authors are after here.}
>
> We have chosen this example because it provides a real data example of a noisy divergence-free field. The vector field would be divergence-free if the buoys followed the water flow. However, the buoys do not precisely follow the water flow, introducing noise. Moreover, we show that enforcing the divergence-free constraint improves the predictions compared to not enforcing it

---

### Review · Reviewer_TGRp · 2025-04-03

**Summary Of Contributions:**

This work proposes MOCK, a multivariate occupation kernel algorithm, to learn the systems of ODEs from trajectories or their snapshots. Comparing to existing quadratic methods, MOCK makes use of the implicit formulation of the vector-valued reproducing kernel Hilbert spaces (vvRKHS) and scales linearly w.r.t. the dimension of state space. Experimental results over synthetic and real-world datasets show promising performance of MOCK. A divergence-free version of MOCK is also proposed and evaluated to show its ability to learn constrained vector fields.

**Audience:**

Yes

**Broader Impact Concerns:**

No broader impact concerns

**Claims And Evidence:**

Yes

**Requested Changes:**

I would like to suggest the authors improve the formatting, for example:

- There are 90 equations in total (which are highly appreciated), but just a few of them are referred in the main paper. Is it better to only show labels for important ones so that readers can capture main results?

- References to equations, figures, and tables are inconsistent, just name a few, equation xx/(xx), f/Figure xx, t/Table XXX, Alg/Algorithm XXX. Could the authors please check the paper again to make those references consistent?

- For referred papers, please use citep and citet properly, e.g., ResNet (He et al. 2016) instead of ResNet He et al. (2016).

- MOCK is a nice name and is used to represent the proposed methods in the whole paper. However, in Table 2, the MOCK methods are replaced by ockG, ockL, etc. to the specific scalar kernel, which makes sense but such inconsistency is a little bit confusing. I suggest the authors either (i) explain ockG etc. in the main context, e.g., Section 4.3 or 4.4, or (ii) use MOCK-G, MOCK-L, etc.

- Since the paper has mainly compared with some of the featured papers for comparison, having a discussion of more recent related works would be benificial for a better position of this paper, more specifically:
    - Using kernel methods within Gaussian Processes for ODE have also been investigated ([1, 2])
    - There are also more recent works in latent ODE approaches ([3, 4, 5])
   Such aspects could be simply discussed in 4.4 for simplicity.

- [1] https://proceedings.mlr.press/v80/heinonen18a/heinonen18a.pdf
- [2] https://proceedings.mlr.press/v180/hegde22a/hegde22a.pdf
- [3] https://arxiv.org/abs/1905.10994
- [4] https://arxiv.org/abs/2103.12413
- [5] https://arxiv.org/abs/2406.02352

**Strengths And Weaknesses:**

This paper is well-written and easy to follow, detailed theoretical derivations are provided. The proposed methods are thoroughly evaluated over various benchmarks against comparable methods. Sufficient discussions and details about these methods are reported, including a complexity comparison reflecting the computational advantage of MOCK. From my perspective, this paper's claim is well supported.

The weakness is mainly from the debatable assumption of I-separable itself, which is crucial for the computational convenience that has been elaborated in this paper, but might at the same time restrictive itself.

---

> ### Author Response · Authors · 2025-04-17
> **Answers to the requested changes and weaknesses**
>
> {\em There are 90 equations in total (which are highly appreciated), but just a few of them are referred in the main paper. Is it better to only show labels for important ones so that readers can capture main results?}
>
> It is our habit to number each equation to facilitate the discussion. We could keep it as is, highlight the most important ones or choose to number only the cited ones. We let the editor weigh in on this.
>
> {\em References to equations, figures, and tables are inconsistent, just name a few, equation xx/(xx), f/Figure xx, t/Table XXX, Alg/Algorithm XXX. Could the authors please check the paper again to make those references consistent?
>
> For referred papers, please use citep and citet properly, e.g., ResNet (He et al. 2016) instead of ResNet He et al. (2016).
>
> MOCK is a nice name and is used to represent the proposed methods in the whole paper. However, in Table 2, the MOCK methods are replaced by ockG, ockL, etc. to the specific scalar kernel, which makes sense but such inconsistency is a little bit confusing. I suggest the authors either (i) explain ockG etc. in the main context, e.g., Section 4.3 or 4.4, or (ii) use MOCK-G, MOCK-L, etc.}
>
> Thank you for pointing out these typos. We will correct them and any remaining typos in the next version of this write-up.
>
> {\em Since the paper has mainly compared with some of the featured papers for comparison, having a discussion of more recent related works would be benificial for a better position of this paper, more specifically:
>
> Using kernel methods within Gaussian Processes for ODE have also been investigated ([1, 2])
> There are also more recent works in latent ODE approaches ([3, 4, 5]) Such aspects could be simply discussed in 4.4 for simplicity.
>
>
> [1] https://proceedings.mlr.press/v80/heinonen18a/heinonen18a.pdf
>
> [2] https://proceedings.mlr.press/v180/hegde22a/hegde22a.pdf
>
> [3] https://arxiv.org/abs/1905.10994
>
> [4] https://arxiv.org/abs/2103.12413
>
> [5] https://arxiv.org/abs/2406.02352
> }
>
> [1] and [2] In both cases, the method presented parametrizes the estimated vector field with a zero-mean Gaussian Process (GP) whose covariance is given by a positive-definite kernel.  A grid of inducing points is formed where the vector field is learned directly, and the full vector field is then interpolated from these inducing points.  To fit the model, the authors maximize the likelihood of generated trajectories at the times corresponding to data, assuming iid additive noise in the observations in the training set.  The use of the grid of inducing points makes this model prohibitively expensive in large dimensions as the number of grid points required increases exponentially with dimension, and thus we have excluded it from our comparison. However, we will cite these paper in the next version of this writeup.
>
> [3] It is about learning a second-order ODE, an interesting problem not addressed in our manuscript.
>
> [4] This method produces a distribution over candidate vector fields that fit the data, acknowledging that oftentimes, many different models are adequate for a given dataset.  The method learns a distribution over initial conditions in latent space and a distribution over control parameters that are input into a latent neural ODE.  The latent ODE is then integrated and decoded when predictions are desired.  The two learned distributions account for the stochasticity in the vector field. This is certainly an intriguing method; however, comparing this method to ours presents challenges, as the two methods solve different problems.  Comparing directly to a neural ODE method seems more appropriate for our purposes, as we have done with LatentODE.
>
> 5] The abstract reads: "consider the problem of optimizing initial conditions and termination time in dynamical systems governed by unknown ordinary differential equations (ODEs)" This paper is not directly relevant since it does not consider the problem of learning a vector field.
>
> [\em The weakness is mainly from the debatable assumption of I-separable itself, which is crucial for the computational convenience that has been elaborated in this paper, but might at the same time restrictive itself.}
>
> Note here that if the kernel is separable, which includes the I-separable case, then computational gain can be obtained. In this case, the vvkernel is of the form K(x,y)=k(x,y)A, where k(x,y) is a scalar kernel and A is PSD. A rotation of the data according to the eigenvectors of A then allows for the same computational gains as for the I-separable kernel, i.e., A=Identity.
> In the case of a divergence-free kernel, the same technique cannot be applied. However, various approximation techniques, e.g., random Fourier features, are still available.

---

### Review · Reviewer_WSpE · 2025-05-22

**Summary Of Contributions:**

Compared to the previous quadratic methods, the author proposed a linear approach to learn the dynamic systems derived from ODEs. The proposed MOCK is easy to implemented and high-performed on the high-dimensional data.

**Audience:**

Yes

**Broader Impact Concerns:**

No.

**Claims And Evidence:**

Yes

**Requested Changes:**

When solving a system of ODEs, it is necessary to discuss the necessity of high-dimensional data or why high-dimensional data often exists. From the experiment, the extremely high dimensions come from the synthetic data set. If the existence of high-dimensional data is explained at the beginning, it can make the paper more well-motivated.

**Strengths And Weaknesses:**

Pros:
The paper has a detailed literature review, propose an easy-to-follow algorithm. The proposed method also has a convincing performance and low computation complexity than other baselines.

Cons:
The method is well-proposed, but the motivation of proposing the method seems not to be adequately discussed.

---

> ### Author Response · Authors · 2025-05-30
> **point well taken**
>
> This is a point well taken.  We propose to add the following paragraphs to the introduction.
>
> Large systems of ODEs may arise from the numerical discretization of a PDE. They also occur in modeling the temporal evolution of complex systems composed of many interacting components. These systems appear across many disciplines.
> We provide some examples.
> \begin{enumerate}
> \item \textbf{The Method of Lines:} Consider a PDE of the form
>
>     $$\frac{\partial}{\partial t}u(x,t) = \mathcal{L}u(x,t)$$
>
> where $\mathcal{L}$ is a differential operator acting on the spatial variable(s). Examples include Burgers' equation, the heat equation, Fisher's equation, and other reaction-diffusion systems. The method of lines \citep{schiesser2012numerical} is a technique for solving PDEs of the form (\ref{eq:example pde}) by discretizing the equation in all but the time variable, reducing it to a system of coupled ODEs (or, possibly, differential algebraic equations). The method proceeds as follows: upon discretizing this system by
> $U_i \approx u(X_i,t)$,$i=1 \ldots d$, we obtain the family of ODEs $dU/dt = G(U)$ where $U$ is a vector of dimension $d$ and $G$ is a $\mathbb{R}^d \to \mathbb{R}^d$ (possibly nonlinear) vector field. If the discretization has, say, 100 points along each of 3 spatial dimensions, this discretization leads to a system of ODEs of dimension one million.
>
> Note that the specific form of $G$ depends on the underlying dynamics and on the spatial discretization scheme. In this paper, we only assume that $G$ belongs to a vvRKHS, but we do not assume a specific parametric form, which is of paramount importance in case $G$ was not fully specified. A setting of special interest is when we have access to trajectory snapshots of the system but not the details on how these snapshots were generated, e.g., the space and/or time discretization parameters.
>
> \item \textbf{Chemical Reaction Networks and Agent-Based Models:} In biochemical systems, each species (e.g., a protein, ion, or small molecule) may follow a distinct concentration trajectory governed by mass-action kinetics or Michaelis-Menten dynamics. The average behavior dynamics often result in a high-dimensional ODE system representing the time evolution of all species in the network. More broadly, agent-based models that describe individual entities (e.g., cells, molecules, or agents) interacting via prescribed rules can often be approximated by high-dimensional ODEs ( \cite{anderson2015stochastic}, \cite{murray2007mathematical}, Chapter 6).
>
> \item \textbf{ Networks of Neurons}: Neural populations can be modeled as networks of coupled nonlinear oscillators, such as the FitzHugh-Nagumo FHN model for individual neuron dynamics. When modeling a large number of neurons with electrical or chemical coupling, one obtains a high-dimensional ODE system (\cite{murray2007mathematical}, Chapter 7).
>
> \item \textbf{Protein Accumulation in the Brain and Neurodegenerative Dynamics:} Models of protein aggregation and propagation in diseases like Alzheimer's or Parkinson’s involve spatial-temporal accumulation and diffusion processes. When discretized in space (e.g., across brain regions in Positron Emission Tomography imaging studies), these systems lead to high-dimensional ODEs for tracking the progression of misfolded proteins (e.g., tau, beta-amyloid).
>
> \item  \textbf{Mechanical Systems with Multiple Degrees of Freedom (Lagrangian and Hamiltonian Formulations):} In robotics and biomechanics, systems often have many degrees of freedom—e.g., each joint or actuator contributes to the state space. The dynamics of such systems are derived from Newtonian, Lagrangian, or Hamiltonian mechanics, yielding structured high-dimensional ODEs (\cite{featherstone2014rigid}, Chapter 7, \cite{ivancevic2010dynamics}).
> \end{enumerate}
>
> References:
>
> David F Anderson and Thomas G Kurtz. Stochastic analysis of biochemical systems, volume 674. Springer,
> 2015.
>
> F. Brauer, P. Van den Driessche, J. Wu, and L. J. S. Allen. Mathematical Epidemiology. Springer, 2008.
> Roy Featherstone. Rigid body dynamics algorithms. Springer, 2014.
>
> Vladimir G Ivancevic and Tijana T Ivancevic. Dynamics and control of humanoid robots: A geometrical
> approach. Paladyn, Journal of Behavioral Robotics, 1(4):204–218, 2010.
>
> J.H. Matis and T.E. Wehrly. Compartmental models of ecological and environmental systems. Handbook of
> Statistics, 12:583–613, 1994.
>
> James D Murray. Mathematical biology: I. An introduction, volume 17. Springer Science & Business Media,
> 2007.
>
> S. A. Peters. Physiologically-based pharmacokinetic (PBPK) modeling and simulations. John Wiley & Sons,
> second edition, 2021.
>
> William E Schiesser. The numerical method of lines: integration of partial differential equations. Elsevier,
> 2012.

---

> > ### Comment · Reviewer_WSpE · 2025-06-08
> > **Thanks for author's revision**
> >
> > After reviewing the revision, I recommend to accept. It is a well-motivated and well-proposed work.

---

### Decision · Action_Editor_uYNh · 2025-06-22

**Recommendation:** Accept as is

**Additional Comments:**

During the rebuttal process, the authors provided detailed clarifications on the theoretical formulation, computational complexity, hyperparameter tuning, and experimental setup. These clarifications were incorporated into the revised manuscript, which also introduced new ablation studies and expanded the discussion on real-world applicability and limitations. Following the revision, all reviewers acknowledged the improvements in clarity, technical rigor, and empirical validation. At this stage, no major concerns remain.

**Audience:**

Yes

**Audience Explanation:**

ODE-based models are becoming increasingly popular for time series analysis and distribution matching (e.g., flow matching). The proposed method for learning nonparametric ODE systems achieves linear scaling with respect to the dimensionality of the state space. This scalability makes it relevant to the TMLR audience, especially those working on high-dimensional dynamical systems, flow-based modeling, and machine learning applications in scientific discovery.

**Claims And Evidence:**

Yes

**Claims Explanation:**

The paper proposes a new algorithm for learning nonparametric systems of ordinary differential equations (ODEs). The theoretical development is rigorous and the exposition has improved significantly in the revised version. Overall, the major claims are well supported and backed by analytical results and many experiments on both synthetic and real-world datasets.

---

> ### Author Response · Authors · 2025-07-18
> **Camera Ready version**
>
> Dear Action Editor,
> Thank you for your work and the work of the reviewers.
> We greatly appreciate the time spent in carefully reviewing and helping to improve our paper during the review process.
> FYI, the camera-ready version has been submitted.
> Best regards,